# Direct photo-patterning of halide perovskites toward machine-learning-assisted erasable photonic cryptography

Yingjie Zhao [1,7], Mengru Zhang[1,7], Zhaokai Wang[1], Haoran Li[1], Yi Hao[1], Yu Chen[2], Lei Jiang [3,4,5], Yuchen Wu [3,4,5] ✉, Shuang-Quan Zang [1] ✉ & Yanlin Song [1,6] ✉

The patterning of perovskites is significant for optical encryption, display, and optoelectronic integrated devices. However, stringent and complex fabrication processes restrict its development and applications. Here, we propose a conceptual methodology to realize erasable patterns based on binary mix-halide perovskite films via a direct photo-patterning technique. Controllable ion migration and photochemical degradation mechanism of iodine-rich regions ensure high-fidelity photoluminescence images with different patterns, sizes, and fast self-erasure time within 5 seconds, yielding erasable photonic cryptography chip, which guarantees the efficient transmission of confidential information and avoids the secondary leakage of information. The ultrafast information encryption, decryption, and erasable processes are attributed to the modulation of the crystallographic orientation of the perovskite film, which lowers the ion migration activation energy and accelerates the ion migration rate. Neural network-assisted multi-level pattern encoding technology with high accuracy and efficiency further enriches the content of the transmitted information and increases the security of the information. This pioneering work provides a strategy and opportunity for the integration of erasable photonic patterning devices based on perovskite materials.

Patterning technology of semiconductors based on the top-down strategy has propelled significant advancement of optoelectronic integrated devices, such as light-emitting diodes[1,2], field-effect transistors[3–5], and photodetectors[6,7]. However, the top-down strategy is incompatible with the patterning of solution-processed semiconductor materials, such as quantum dots, organic semiconductors, and halide perovskites, which can damage the physical and chemical properties of the material,

sacrificing the optoelectronic performance of the devices[8–11]. For the patterning of such materials, a series of solution processing methods have been developed, such as inkjet printing, stencil-assisted printing, and dip-pen printing[12–14]. Among them, perovskite materials have been exploited in the field of patterned optoelectronic devices owing to their excellent optoelectronic performance, solution processing, diverse crystal structures, and high photoluminescence quantum yield, but they still suffer from

[1]College of Chemistry and Pingyuan Laboratory, Zhengzhou University, Zhengzhou 450001, PR China. [2]The Institute of High Energy Physics, Chinese Academy of Sciences, Beijing 100049, PR China. [3]Key Laboratory of Bio-inspired Materials and Interfacial Science, Technical Institute of Physics and Chemistry, Chinese Academy of Sciences, Beijing 100190, PR China. [4]Suzhou Institute for Advanced Research, University of Science and Technology of China, Suzhou 215123, China. [5]University of Chinese Academy of Sciences (UCAS), Beijing 100049, China. [6]Key Laboratory of Green Printing, Institute of Chemistry, Chinese Academy of Sciences, Beijing 100190, PR China. [7]These authors contributed equally: Yingjie Zhao, Mengru Zhang.
✉e-mail: wuyuchen@iccas.ac.cn; zangsqzg@zzu.edu.cn; ylsong@iccas.ac.cn

trade-offs in low processing efficiency, nonuniformity, complex processes, and high costs, significantly restricting their practical application[15–24].

Lately, the direct laser writing (DLW) technique was successfully applied to the patterning of perovskite materials, realizing the fabrication of functional optoelectronic devices[25–29]. For example, Gan and his colleagues realized multicolor fluorescence patterns based on the DLW technique, which provides insights into anticounterfeiting and steganography[30]. Furthermore, direct photolithography has also been developed for patterning perovskite quantum dots with photosensitive ligands, which can crosslink and thus change the solubility of the quantum dots, providing an efficient and uniform patterning strategy[31–33]. Nevertheless, the synthesis of quantum dots usually undergoes a complicated process of high temperature and inert atmosphere protection, and its ionic crystal nature, unstable surface-bonded ligands, tends to produce defects and sacrifice its photoluminescence properties[34]. In addition, high-resolution patterned photolithography processes for quantum dots usually irreversibly degrade the photophysical properties of the material and use environmentally unfriendly solvents, further limiting its commercialization[32]. Therefore, it is urgent to develop an ideal microscale patterning method for low-cost, high-throughput, high-resolution patterning fabrication of perovskite materials.

In this work, we first realize erasable photonic patterns with different compositions, colors, shapes, and sizes based on binary mix-halide perovskite films via a direct photo-patterning technique, yielding erasable photonic pattern encryption and multilevel pattern encoding technology (Fig. 1a–c). The realization of photonic pattern encryption information with clear outlines and boundaries is mainly attributed to the controlled phase separation caused by halide ion migration and the photochemical degradation of iodine-rich regions, yielding the photoluminescence difference between the exposed and unexposed regions (Fig. 1d). Self-erasure of encrypted information occurs simultaneously with the decryption of the information, which is attributed to the rapid ion migration and photodegradation modulated by crystallographic orientation. The self-erasure time within 5 s can effectively prevent secondary leakage of information and ensure information security. The unusual spectral blue shift phenomenon arises from phase separation and photodegradation of the iodine-rich regions. Furthermore, neural network-assisted multi-level photonic pattern encoding and decoding with higher accuracy and efficiency are demonstrated, which further enriches the information content and increases the security of the information based on photonic cryptography (Fig. 1e, f). This work provides insight into low-cost and high-speed self-erasure encryption information technology based on perovskites.

## Results

### Direct photo-patterning technique
The simple, low-cost, high-throughput, and high-precision patterning method is urgently needed for perovskite materials, but fabrication remains a major challenge. To address these issues, an unprecedented patterning technique with a self-erasure function was demonstrated via controlled ion migration within binary mix-halide perovskite films based on the direct photo-patterning technique (Fig. 1a, b). The realization of photon patterns relies on the introduction of a photo mask, where the hollowed areas allow the transmission of ultraviolet (UV) light, resulting in the migration of halide ions, and the non-hollowed areas do not allow the transmission of UV light, preventing the migration of halide ions, eventually realizing the photonic pattern. Patterned information is read under UV illumination, while its pattern information is hidden in the bright field image. The patterned information is erased simultaneously with the reading of the information, thus avoiding the secondary leakage of information (Fig. 1a, c, and Supplementary Fig. 1). The photochemical reaction mechanism of

patterning is discussed in detail below. The advantages of the self-erasure photonic patterning technique include three aspects. Firstly, the low-cost solution processing and fast photonic pattern are highly compatible with rigid and flexible substrates, favoring the rapid production of patterned photonic devices. Secondly, the self-erasure photonic patterning technique is universally compatible with various types of mixed-halide perovskite films, enabling the realization of self-erasure photonic patterning with different perovskite compositions, shapes, colors, and sizes. Thirdly, the fast self-erasure process and the encoding of photonic patterns enable the encryption of information, which significantly improves information security and increases information capacity, providing a field for photonic devices based on perovskites. The realization of the pattern information encryption is mainly attributed to the controlled ion migration and the photodegradation of the iodine-rich regions within the binary mix-halide perovskite films, resulting in the formation and erasure of the encrypted information.

### Crystal facet dependence of ion migration
The fabrication of photonic patterns mainly relies on controlled ion migration and degradation of iodine-rich regions within the binary mix-halide perovskite films based on the direct photo-patterning technique. Under UV illumination, phase separation resulting from rapid ion migration facilitates fast pattern writing and self-erasure, yielding high-performance photonic cryptographic chips. The different crystal facets of perovskite usually correspond to different ion migration velocities due to the different halide ion mobility activation energy[35,36]. To evaluate the effect of different crystal facets on halide ion migration, the perovskite films with different crystal facets were fabricated through additive engineering with 1-butyl-3-methylimidazolium bis(trifluoromethylsulfonyl)imide (BMITFSI) (see Methods section for details). Photoluminescence spectra of perovskite films processed with and without BMITFSI additive exhibit small differences (Supplementary Fig. 2). Figure 2a shows the X-ray diffraction (XRD) patterns of perovskite films processed with different ratios of additives. In contrast to the (100)-dominant crystallographic orientation using pure solvent, the perovskite films present stronger (110) diffraction peaks with the introduction of the BMITFSI additive, indicating the successful tailoring of the crystal facets. With the increased proportion of additives, the perovskite films present the (110)-dominant crystallographic orientation, accompanied by an increased full width at half-maximum (fwhm), which can be attributed to a reduced grain size of perovskite (Supplementary Fig. 3). Tailoring of crystal facets using different ratios of additives is further demonstrated by grazing-incidence wide-angle X-ray scattering (GIWAXS) results, in which the stronger Bragg diffraction spot assigned to (110) crystal facets occurs using additives (Fig. 2b–d). The modulation of the crystallographic orientation of the perovskite films can be ascribed to the fact that the BMITFSI additive is more adsorbed onto the 110 crystal facets of the perovskite, thus suppressing its growth[37,38]. Figure 2e and f show the dominant crystal facets processed by using pure solvent and additive, respectively. Fabrication of photonic patterns requires dense and uniform perovskite films. The morphology of perovskite films is revealed by scanning electron microscopy (SEM), optical microscopy, and fluorescence microscopy. For the pure solvent, the perovskite films showed discrete island-like morphology with large grain sizes, which is not favorable for the production of photonic patterns. In contrast, perovskite films fabricated by the introduction of additives show a denser film morphology with a smaller grain size, which is consistent with XRD results (Fig. 2g, h, and Supplementary Fig. 4).

Then, we investigated the ion migration velocity of different crystal facets in binary mix-halide perovskite films by monitoring the photoluminescence spectral evolution under UV illumination and measuring the steady-state I-V curves. Compared to pure bromine

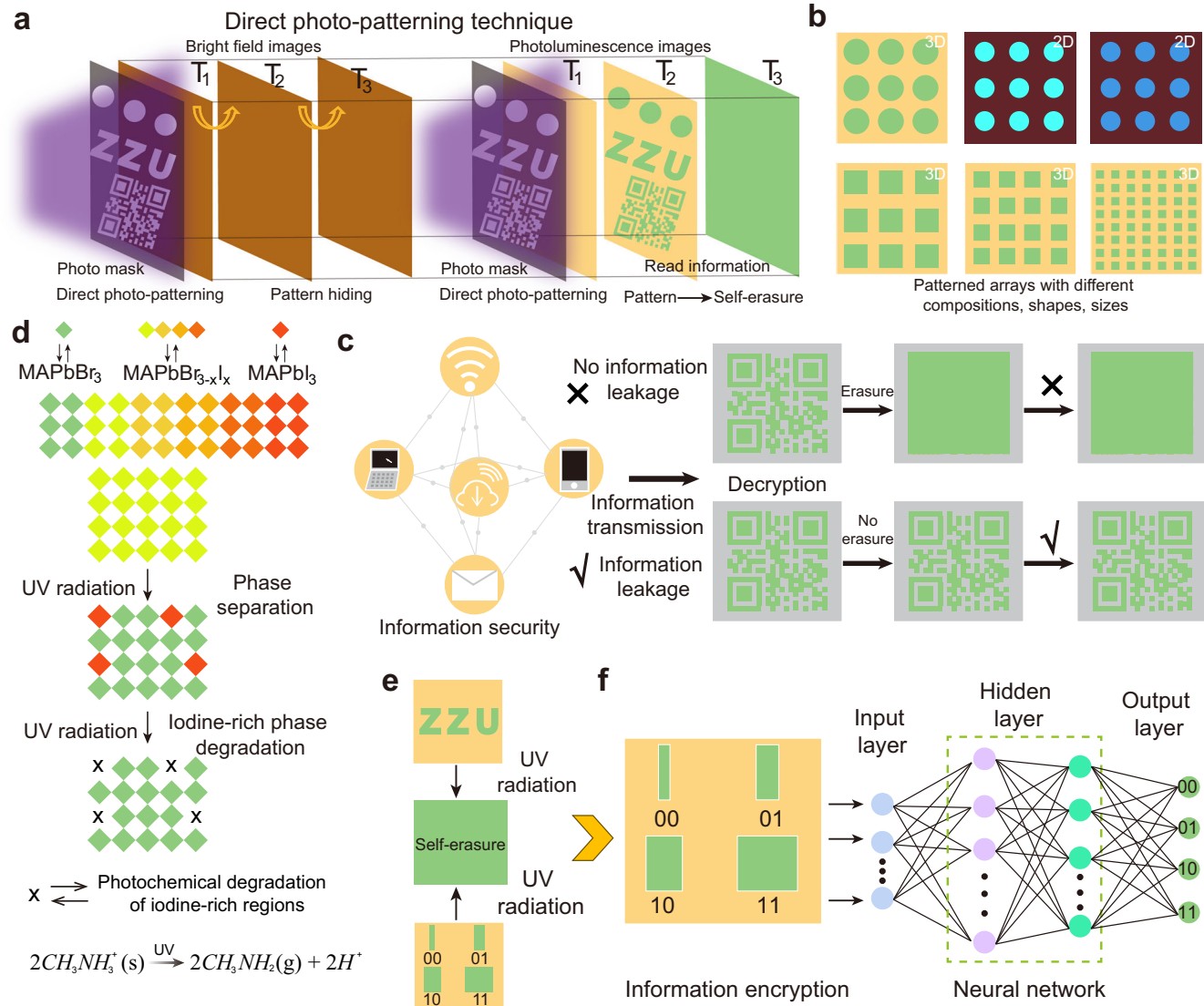

**Fig. 1 | Design principle and application of direct photo-patterning technique.**
**a** Schematic diagram and photonic pattern based on direct photo-patterning tech-nique. Self-erasure of pattern information simultaneously occurs with the read information. $T_1$, $T_2$, and $T_3$ represent different times of the direct photo-patterning process, $T_1 < T_2 < T_3$. **b** The universal pattern fabrication of direct photo-patterning technique for different perovskite compositions, shapes, and sizes. The yellow and crimson background photoluminescence colors correspond to binary mix-halide three-dimensional (3D) and two-dimensional (2D) perovskite films, respectively.
**c** The importance of information transmission security. The main information transmission methods currently include letters, computers, mobile phones, and

other types of wireless transmission devices. The real information of the quick response (QR) code is the letters "ZZU". **d** Mechanism diagram of direct photo-patterning technique by pattern-controlled ion migration and photodegradation of the iodine-rich regions within mix-halide perovskite films. The red and green octa-hedra represent $MAPbI_3$ and $MAPbBr_3$ perovskites, respectively. The octahedra with other colors represent mixed bromine-iodine perovskites. **e** Schematic diagram of photonic pattern encryption and multilevel pattern encoding based on direct photo-printing technique. **f** Neural network-assisted information decryption. Neural net-work simulation enables highly accurate recognition of the encrypted pattern sizes corresponding to different binary codes, thus decrypting the information.

perovskite film, the diffraction peaks of bromine-iodine-doped per-ovskite films are blue-shifted, indicating the success of ion exchange (Supplementary Fig. 5). Furthermore, the denser film morphology and the same tailoring effect of the crystal facets has also been demon-strated in binary mix-halide perovskite films by the introduction of BMITFSI additives (Supplementary Figs. 6–9). Under continuous UV illumination, the mix-halide perovskite films with (110)-dominant crystallographic orientations show faster color changes, indicating a faster ion migration rate (Fig. 2i, j, and Supplementary Fig. 10). Ion migration usually leads to large hysteresis phenomenon for the *I-V* curves[39,40]. Supplementary Fig. 11 shows the logarithmic *I-V* curves under forward sweep and reverse sweep, and the films with (110)-dominant crystallographic orientations show a larger hysteresis, fur-ther illustrating the more severe ionic migration phenomenon. To

explain the differences in ion migration rates for different crystal facets, we systematically measured the ion migration activation energy of perovskite single-crystal with different crystal facets by the temperature-dependent conductivity measurements (details shown in Supplementary Figs. 12–15 and Supplementary Notes S1, S2)[41,42]. Compared to the *E*a value of 0.31 eV of the (100) crystal facet, a lower *E*a value of 0.16 eV of the (110) crystal facet was demonstrated, which is close to previously reported literature[43,44]. Furthermore, the ion migration activation energy on the different crystal facets were cal-culated based on DFT calculation, which is well in agreement with the experimental value (Fig. 2k–n, and Supplementary Fig. 16). Overall, in contrast to the (100) crystal facets, the (110) crystal facets have a smaller ion migration activation energy, thus leading to their higher ion migration rate (Fig. 2m, n).

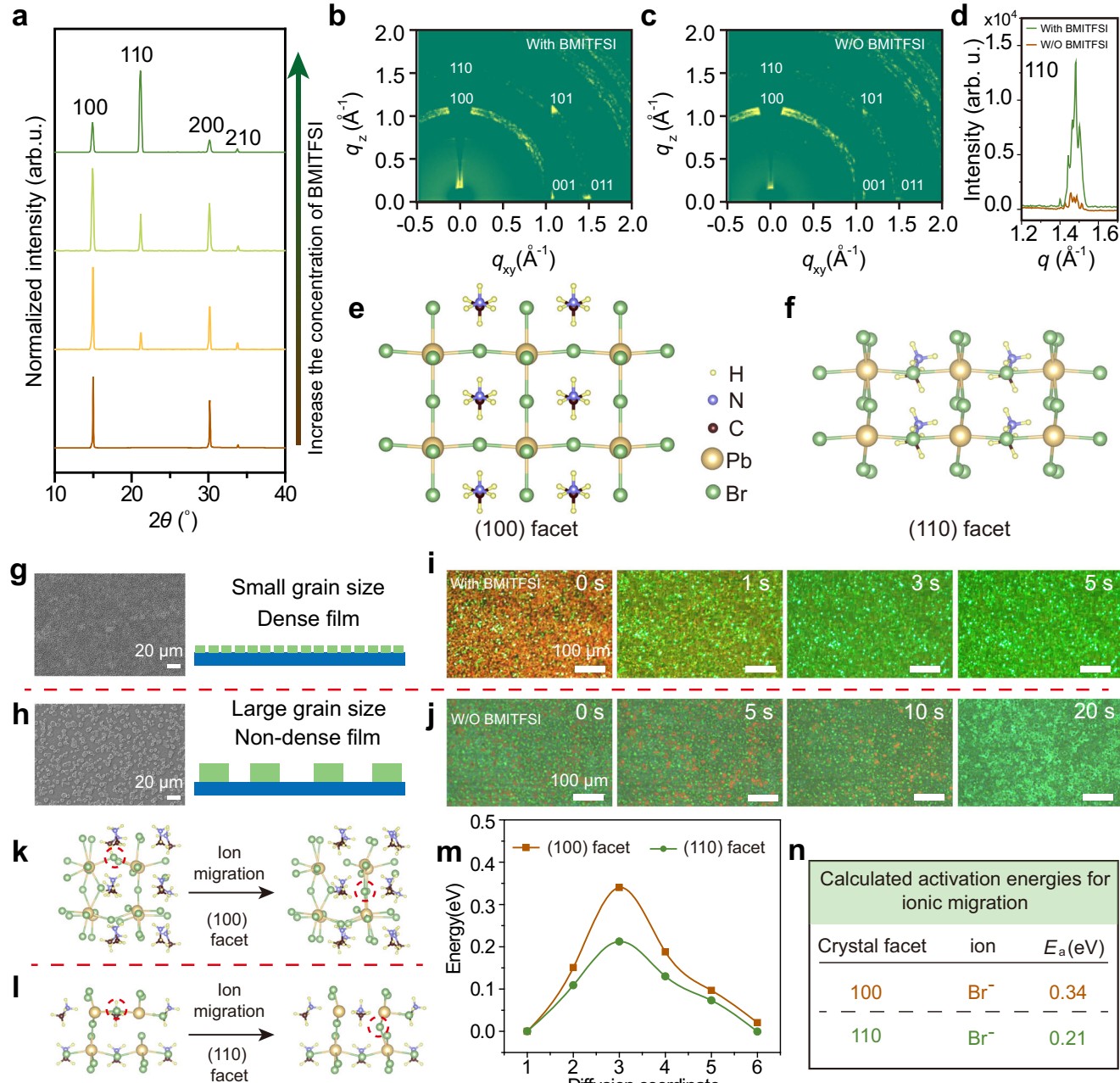

**Fig. 2 | Crystal facet-dependent ion migration of MAPbBr₃ perovskite films.**
**a** X-ray diffraction (XRD) pattern of perovskite films with pure solvent and different ratios of 1-butyl-3-methylimidazolium bis(trifluoromethylsulfonyl)imide (BMITFSI) additives. Grazing-incidence wide-angle X-ray scattering (GIWAXS) patterns of perovskite films (**b**), with additive, and (**c**) without additive. **d** Diffraction peak intensity of (110) crystal facets of perovskite films with additive and without additive. The schematic diagram of the (**e**), (100) crystal facet, and the (**f**), (110) crystal facet. Scanning electron microscopy (SEM) morphology and the corresponding schematic of perovskite films fabricated (**g**), with additive and (**h**), without BMITFSI additive. Photoluminescence photographs of mix-halide perovskite films. **i**, with additive, and (**j**), without BMITFSI additive under different ultraviolet (UV) irradiation times. The bromide ion migration paths in the (**k**), (100) crystal facets and the (**l**), (110) crystal facets. The red dashed circles represent the location of migrating ion. **m, n** The calculated ion migration activation energies for the (100) crystal facets and the (110) crystal facets.

## Photochemical reaction mechanism of patterning

Given that the MAPbBr₃ perovskite films with (110)-dominant crystallographic orientations have higher ion migration ability, thus we choose them as the component of the photonic patterns. Then, we systematically analyzed the photochemical reaction mechanism of the patterning. To investigate the effect of methylammonium iodide (MAI) concentrations on the binary mix-halide perovskite films, absorption, and photoluminescence spectrum with different MAI doping concentrations under UV irradiation were monitored. The absorption and the photoluminescence spectrum of the perovskite film undergoes a significant redshift with the increased doping concentration of the MAI, which can be attributed to the introduction of iodide ions, reducing the bandgap of the perovskite film (Fig. 3a, b). The Commission Internationale de L' Eclairage (CIE) coordinate plot shows more clearly the change of photoluminescence spectrum under UV irradiation (Fig. 3c). The corresponding photoluminescence photographs also showed a noticeable red shift with the increased MAI concentration, indicating the success of ion exchange (Supplementary

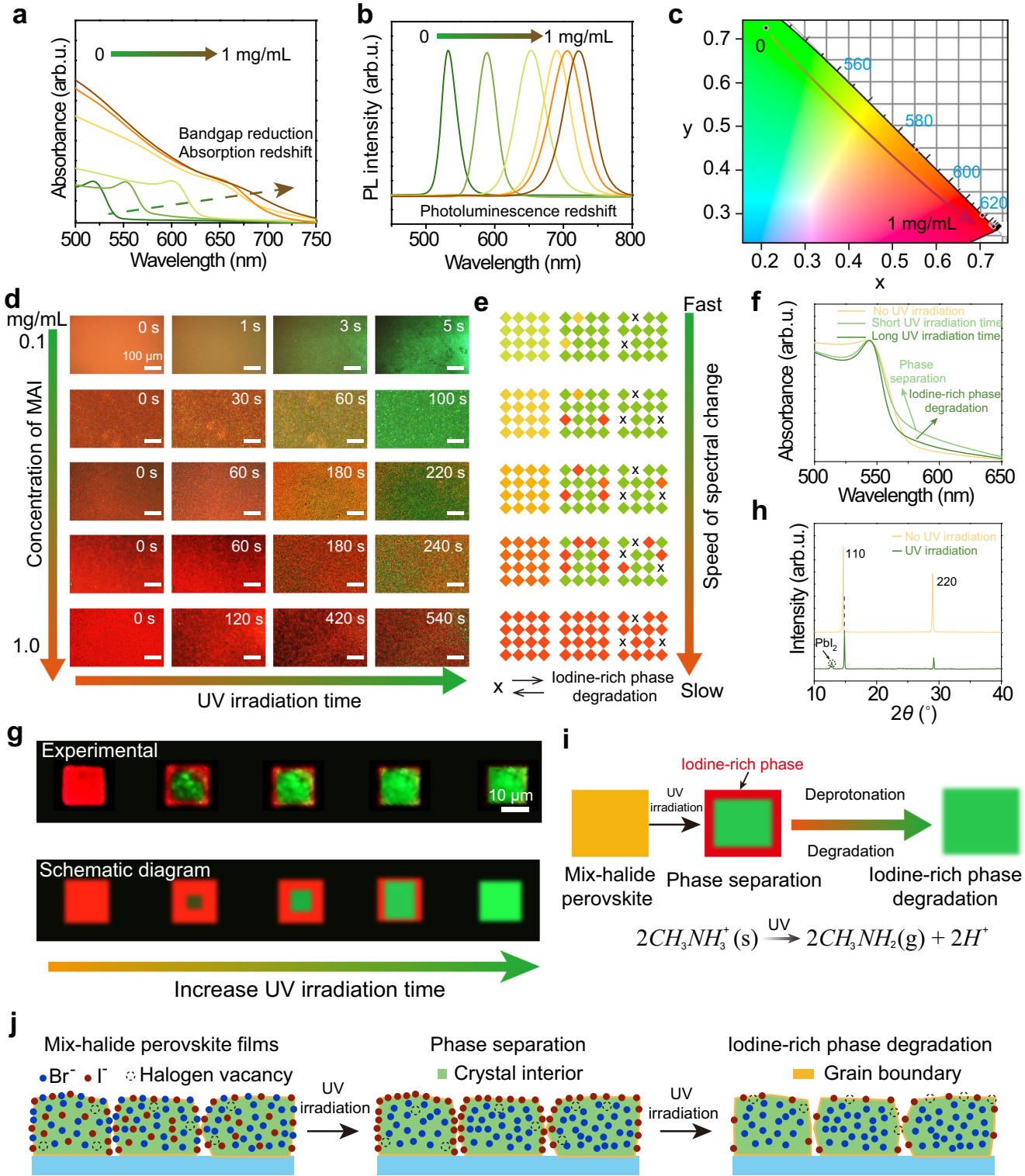

**Fig. 3 | Photochemical reaction mechanism of patterning based on mix-halide perovskite films. a** The absorption, and **b** the photoluminescence spectrum of the MAPbBr₃ perovskite film with the different methylammonium iodide (MAI) doping concentrations. **c** Commission Internationale de L' Eclairage (CIE) coordinates of mix-halide perovskite films with the different MAI doping concentrations.
**d** Photoluminescence photographs of mix-halide perovskite films with different MAI doping concentrations under different ultraviolet (UV) irradiation times. **e** The mechanism of the photoluminescence variation of mix-halide perovskite films with different MAI doping concentrations before and after UV irradiation. **f** Absorption spectra of the mix-halide perovskite films with MAI concentration of 0.1 mg/ml at

different UV irradiation times. **g** Photoluminescence photographs and schematic diagram of mix-halide perovskite single crystals under different UV irradiation times, revealing phase separation and decomposition of iodine-rich phases. The top picture is the experimental data, and the bottom picture is the corresponding schematic diagram. **h** X-ray diffraction (XRD) pattern of the mix-halide perovskite film before and after UV irradiation. **i** Decomposition mechanism of mix-halide perovskite film under UV irradiation. **j** Microscopic ion distribution during phase separation and decomposition of mix-halide perovskite film. The gradient color arrows are added to increase the readability of Fig. 3 and have no physical properties.

Fig. 17). Furthermore, no obvious morphological difference is observed in the optical photographs before and after the ion exchange, indicating that the ion exchange is a mild and non-destructive process, providing guarantees for patterning (Supplementary Fig. 18).

Figure 3d shows the photoluminescence images of MAPbBr$_3$ perovskite films doped with the different MAI concentrations under different UV irradiation times. At low doping concentrations, the photoluminescence intensity of perovskite films increases accompanied by a blueshift with the increased UV irradiation time, which is mainly attributed to the rapid ionic migration under UV irradiation, leading to the redistribution of the bromine and iodine phases, and hence the spectral change. With the increased doping concentration, the blue shift rate of the perovskite films decreases until no blueshift occurs, mainly because the entire film becomes an iodine-rich phase and thus will not degrade completely (Fig. 3d)[45]. Under the ion doping concentration of 0.1 mg/ml, the fastest photoluminescence blueshift time is less than 5 s, providing a guarantee for fast self-erasure photonic cryptography. Meanwhile, no morphology changes were observed (Supplementary Fig. 19). Generally, the phase separation will result in a red shift of the photoluminescence peaks, however, no red-shifted photoluminescence peaks were found for low MAI concentrations, which can be attributed to the degradation of the iodine-enriched phase in the process of phase separation, and thus the spectra are always blue-shifted[45]. For high MAI concentrations, no blue-shifted photoluminescence is attributed to the inhibition of phase separation and the minor decomposition of the iodine-rich phase, thus there is no difference in the photoluminescence photographs (Fig. 3e). Overall, the photoluminescence spectra of the perovskite films can be controlled by regulating the doping concentration of the MAI salt and the time of UV irradiation for producing high-contrast patterns.

To deeply investigate the origin of the photoluminescence blue shift after phase separation, i.e., the disappearance mechanism of photoluminescence at longer wavelengths, we carried out a thorough analysis. First, the mix-halide perovskite films with an MAI concentration of 0.1 mg/ml were fabricated. Figure 3f shows the evolution of the absorption spectra under different UV illumination times. Under short-term UV irradiation, the enhanced absorption at longer wavelengths is attributed to the generation of iodine-rich phases arising from the onset of phase separation. With the increased UV irradiation time, the absorption intensity at long wavelengths starts to decrease, which can be attributed to the degradation of the iodine-rich region. The decreased photoluminescence intensity at long wavelengths and enhanced photoluminescence at short wavelengths further demonstrate the phase separation and degradation process of mix-halide perovskite films, which is consistent with the absorption results (Supplementary Fig. 20). Because the size of the grains within the polycrystalline film is small, it is difficult to observe the phase separation and decomposition process in the iodine-rich region using in situ optical characterization, so we fabricated mix-halide perovskite single crystals with a size of roughly 10 micrometers (Supplementary Figs. 21 and 22). Under continuous UV irradiation, a clear phase separation process was observed with iodine-rich edge regions and bromine-rich interior regions, which is consistent with reported results in the literature[46]. Furthermore, we found the disappearance of the iodine-rich region with the increased UV irradiation time, which can be attributed to the photodegradation of the iodine-rich phase (Fig. 3g)[45]. Phase separation and decomposition processes were also demonstrated by the photoluminescence spectra (Supplementary Figs. 23 and 24).

Finally, we elucidate the underlying mechanism for the photo-degradation of perovskite films using Fourier Transform Infrared spectra (FTIR) and XRD characterization. As shown in Supplementary Fig. 25, the N-H stretch vibration and C-H bend vibration intensity within the mix-halide perovskite films are noticeably reduced after UV illumination, revealing the deprotonation reaction of the CH$_3$NH$_3^+$ ions, which has been demonstrated in previously literatures[47,48]. The attenuated diffraction peak intensity of (110) crystal facet and diffraction peak appearance of lead iodide after UV irradiation further indicate the photodegradation of mix-halide perovskite films (Fig. 3h). The diffraction peaks shifting to a larger diffraction angle also indicates the decomposition of the iodine-rich phase, which leaves a more stable bromine-rich phase perovskite. Under UV irradiation, the decomposition mechanism and the corresponding microscopic ion distribution of mix-halide perovskite film are shown in Fig. 3i, j. Overall, the decomposition of the mix-halide perovskite film can be attributed to the deprotonation reaction of the CH$_3$NH$_3^+$ ions under UV irradiation, resulting in the complete decomposition of the perovskite structure. Controlled ion migration and photochemical decomposition process within the iodine-rich region providing a guarantee for the realization of high-fidelity, high-throughput patterns.

## Fabrication of photonic pattern and encryption

Given that controlled color transformations can be realized within mix-halide perovskite films, self-erasure photonic pattern encryption was further demonstrated via direct photo-patterning technique (Fig. 4a). Besides 3D mix-halide perovskite films, the direct photo-patterning technique is also applicable to 2D mixed halide perovskite films. Figure 4b demonstrates the universal patterning fabrication based on the direct photo-patterning technique, which enables the realization of patterns with different perovskite compositions, shapes, and sizes. Based on the direct photo-patterning technique, the pattern of the photoluminescence encryption information is recorded into the encryption chip. The decryption information is realized by the irradiation of UV light without a photo mask, which results in rapid halide ion migration over the entire exposure area of the encryption chip, causing the disappearance of the encrypted pattern and realizing the self-erasure function after decryption. The resolution of patterning can reach up to 3175 PPI, which provides the feasibility of high-density data storage (Supplementary Fig. 26). Figure 4c shows the decryption and self-erasure process for the letter "ZZU". The high-fidelity pattern and the fast disappearance of the encrypted pattern within 5 s indicate the feasibility of self-erasure photonic pattern encryption. Encryption, decryption, and self-erasure processes for different-sized patterns and various numerical patterns from 1 to 9 further demonstrate the universality of self-erasure photonic pattern encryption (Fig. 4d and Supplementary Figs. 27 and 28). To demonstrate the reproducible encryption properties of the photonic cryptographic chip, multiple encryptions, decryptions, and self-erasure are carried out, showing excellent reliability, providing a guarantee for the practical application (Fig. 4e, f, Supplementary Figs. 29, 30, and details shown in Supplementary Note S3). It's worth noting that those encryption chips also present remarkable humidity stability, which can be confirmed by the photoluminescence spectrum and the encryption patterns with no obvious difference under different humidity conditions (Supplementary Figs. 31 and 32). To further improve the humidity stability of encryption chips, low-dimensional mixed halide perovskite films can be developed. The direct photo-patterning technique has also been used for binary mix-halide Sn-based perovskite films, providing an environmentally friendly platform (Supplementary Figs. 33 and 34).

## Self-erasure multilevel pattern encoding

Despite the fact that photonic pattern information encryption can transmit basic pattern and letter information, the amount of information transmission is limited and does not have anti-counterfeiting properties. To further enrich the content of the transmitted information, a self-erasure quick response (QR) code and multilevel pattern encoding chips with more information and anti-counterfeiting capability are fabricated. Through decoding information of the

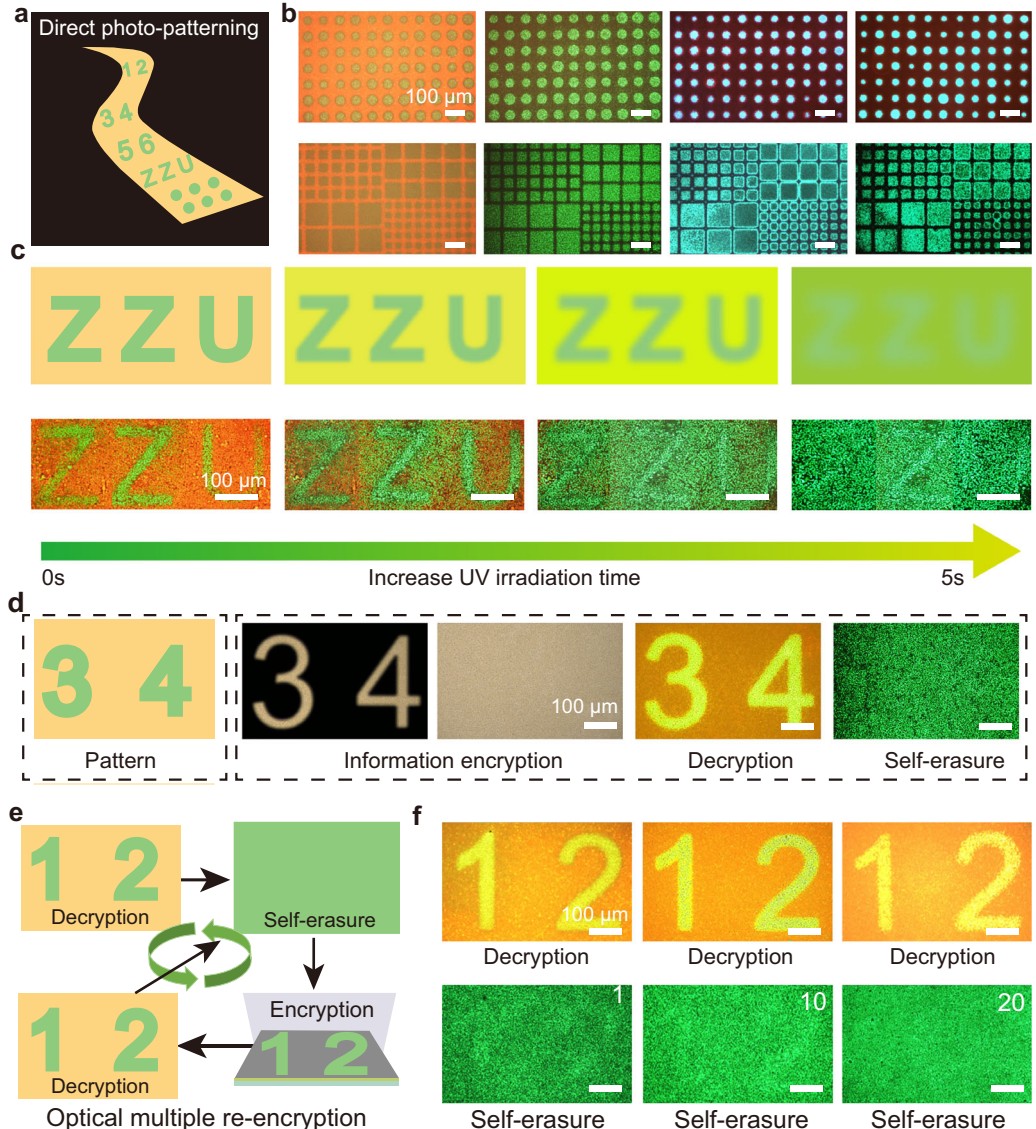

**Fig. 4 | Self-erasure photonic pattern encryption. a** Schematic diagram of photonic pattern encryption based on direct photo-patterning technique. **b** The universal patterning fabrication based on three-dimensional (3D) or two-dimensional (2D) mixed halide perovskite films with different perovskite compositions, shapes, and sizes. **c, d** Schematic and photoluminescence photographs of the decryption and self-erasure process of photonic pattern information. **e, f** Multiple re-encryption of photonic pattern information, confirming excellent reliability. The figures are extracted from Supplementary Fig. 30. The white number represents the cycle times.

encoding pattern, the real information can be further decrypted. Figure 5a shows the schematic diagram of the fabrication of the multilevel pattern encryption chip and the process of decoding information. Figure 5b shows the encryption, pattern decryption, decoding information, and self-erasure process of a photonic pattern QR code containing the real information of "ZZU", demonstrating the feasibility of high-level pattern information encryption. Furthermore, a novel self-erasure photonic coding technique is realized. Supplementary Fig. 35a presents the binary encryption rules for pattern information by strictly controlling the length as well as the width of the photonic pattern. The detailed corresponding relationship between binary and photonic encoding is shown in Supplementary Table 1 and Supplementary Table2. To more clearly show the conversion relationship between the different encoding rules, the corresponding code rule for the letter Z is presented (Supplementary Fig. 35b). The real information "ZZU" was decoded by the photonic pattern information, enabling the prevention of information deciphering and leakage, indicating the excellent security of the self-

erasure photonic information coding (Fig. 5c and Supplementary Figs. 36 and 37). For more accurate and efficient decoding of pattern information, neural network-assisted image recognition is employed. High recognition accuracy ensures accurate decoding of encrypted information (Fig. 5d–f). Furthermore, clear encryption patterns can also be obtained by photolithography and direct-writing lithography, providing multiple fabrication routes for pattern information encryption (Supplementary Figs. 38 and 39). Overall, the self-erasure multilevel pattern encoding chip provides insight into the secure transmission of information.

## Discussion

A simple and efficient strategy for the fabrication of self-erasure photonic cryptographic chips based on perovskite films was demonstrated for the first time via the direct photo-patterning technique. The realization of fast encryption, decryption, and self-erasure of photonic pattern information relies on the controlled ion migration within the binary mix-halide perovskite films, mainly arising from the modulation

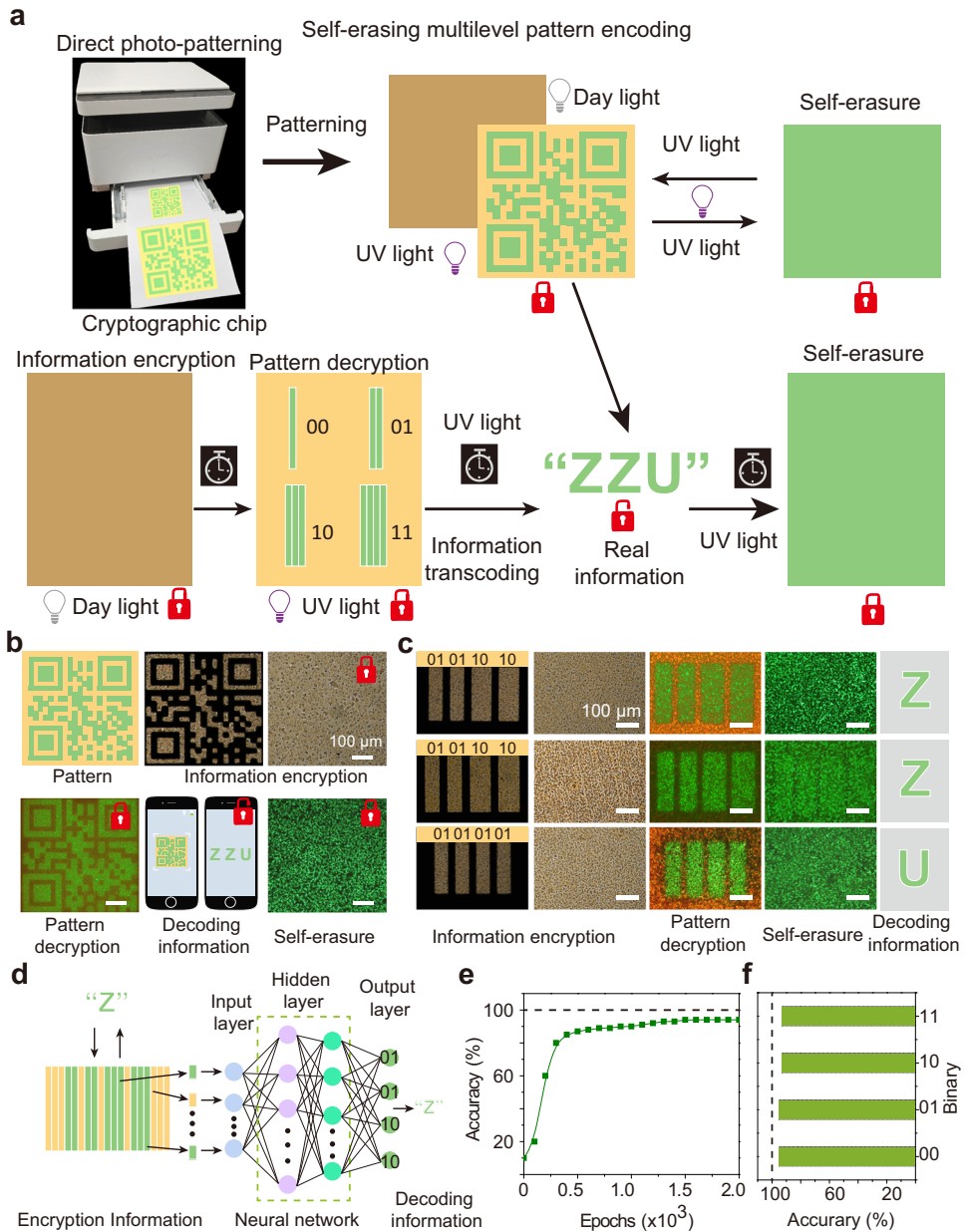

**Fig. 5 | Self-erasure multilevel pattern encoding. a** The design principle of self-erasure multi-level pattern encryption chip. The self-erasure quick response (QR) code and multilevel pattern encoding chips with more information and anti-counterfeiting capability are fabricated. The real information can be further decrypted by pattern under ultraviolet (UV) irradiation before the self-erasure. The real information of the QR code is the letters "ZZU". **b** Photonic pattern encoding based on QR code. **c** Encryption, decryption, and self-erasure process of photonic pattern encoding for "ZZU". **d** Neural network-assisted image recognition for photonic pattern decoding. **e** Pattern recognition accuracy of encryption information during 2000 training epochs. **f** The recognition accuracy of different binary encryption pattern information after 2000 training epochs.

of the crystallographic orientation of the perovskite film, which lowers the ion migration activation energy and accelerates the ion migration rate. Furthermore, neural network-assisted multi-level photonic pattern encoding and decoding with higher storage data capacity and more secure anti-counterfeiting features are realized by customized encoding rules. To further increase the processing efficiency and resolution of photonic pattern chips, lasers with micrometer-sized light spots can be used, enabling the direct encoding of information on the photonic chip in future work. This work provides a research perspective for smart photonic cryptographic chips and opens a paradigm for large-scale, high-throughput erasable pattern fabrication of perovskite films.

## Methods

### Materials

Methylammonium bromide (MABr, 99.9%), Methylammonium iodide (MAI, 99.9%), Phenethylamine iodine (PEAI, 99.9%), Phenethylamine bromide (PEABr, 99.9%) were purchased from Advanced Election Technology CO., Ltd. Lead bromide ($PbBr_2$, 99.99%), Lead iodide ($PbI_2$, 99.99%) were purchased from Xi'an Yuri Solar Co., Ltd. $N$, $N$-dimethylformamide (DMF, ≥ 99.9%) were purchased from Sigma-Aldrich company. Isopropanol ($C_3H_8O$), 1-butyl-3-methylimidazolium bis(trifluoromethylsulfonyl)imide (BMITFSI, 99.5%) were purchased from Innochem company. All chemicals were used without further purification.

## Fabrication of perovskite films

To precise the composition of $MAPbBr_3$ perovskite, perovskite powder was first synthesized. The XRD, absorption, and photoluminescence spectrum indicate phase purity (Supplementary Figs. 40, 41). The perovskite precursor solution (0.5 M) was prepared by dissolving $MAPbBr_3$ perovskite powder into a solution of DMF. For the perovskite film without the BMITFSI additive, the perovskite films were fabricated by spin-coating at 3000 rpm for 45 s and annealing under 70 °C for 5 minutes. For the perovskite film with different ratios of BMITFSI additive (volume ratio relative to perovskite solution), perovskite films were deposited by spin-coating at 3000 rpm for 45 s and annealing under 70 °C for 5 minutes. The $PEA_2PbI_4$ ($PEA_2PbBr_4$) perovskite precursor solution (0.5 M) was prepared by dissolving PEAI (PEABr) and $PbI_2$ ($PbBr_2$) into a solution of DMF, and perovskite films were deposited by spin-coating at 3000 rpm for 45 s and annealing under 100 °C for 15 minutes. For binary mix-halide perovskite films, different concentrations of MAI dissolved in an isopropanol solution were spin-coated onto the surface of the perovskite film for an ion exchange reaction.

## Characterization

The crystallinity of perovskite powder and perovskite films was measured by an X-ray diffractometer (Bruker, D8 focus, Germany). The morphology of perovskite films was measured by SEM (Hitachi, S-8010, Japan). Absorption spectra were obtained by UV-vis-NIR spectrometer (Cary 7000, Agilent, America). PL spectra were acquired by Edinburgh Instruments (FLS1000, England). FTIR spectral curves were acquired by FTIR spectroscopy (Thermo Scientific, Nicolet iS10, America). The GIWAXS data were obtained at 1W1A Diffuse X-ray Scattering Station, Beijing Synchrotron Radiation Facility (BSRF-1W1A). The $I\text{-}V$ curves and the temperature-dependent conductivity were measured using a vacuum manual probe station (Lake Shore) and a 4200 semiconductor characterization system (Keithley, 4200). The device temperature was controlled with a thermoelectric plate and liquid nitrogen. The photoluminescence patterns were recorded using a home-made optical setup.

## Calculation

We used the DFT as implemented in the Vienna Ab initio simulation package (VASP) in all calculations. The exchange-correlation potential is described by using the generalized gradient approximation of Perdew-Burke-Ernzerhof (GGA-PBE). The projector augmented-wave (PAW) method is employed to treat interactions between ion cores and valence electrons. The plane-wave cutoff energy was fixed to 450 eV. Given structural models were relaxed until the Hellmann–Feynman forces were smaller than -0.02 eV/Å and the change in energy smaller than $10^{-5}$ eV was attained. Grimme's DFT-D3 methodology was used to describe the dispersion interactions among all the atoms in adsorption models. For searching the transition states, we employ the nudged elastic band (NEB) method as implemented in VASP.

## Simulation of neural networks

To simplify the neural network simulation, the image containing the encrypted pattern information is first sliced into equally spaced units in the width direction, and no slicing is performed in the length direction, and each of the sliced strips contains a different color. The number of stripes within the green pattern corresponds to different binary codes. The neural networks were composed of the input layer, hidden layer, and output layer. The sigmoid function unit was used as a mathematical activation function in neural networks. The back-propagation algorithm is used for training and recognition processes in our neural network. The learning rate is set as a fixed value of 0.01. The training utilized 2000 training images and 1000 test images dataset at each epoch. After 2000 epochs, the recognition accuracy was calculated for evaluation. The decoding information can finally be obtained by identifying the number of stripes within the green pattern.

## Data availability

All data supporting the findings of this study are available within the article and the Supplementary Information file. The raw data are available via Zenodo at https://doi.org/10.5281/zenodo.14915213. Source data are provided with this paper (ref. 49). Source data are provided with this paper.

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

## Acknowledgements

This work was supported by the National Key R&D Program of China (Grant No. 2023YFE0111500) (Y.S.), the National Natural Science Foun-dation of China (T2425026 (Y.W.), 52303257 (Y.Z.), 52321006 (Y.S.), T2394480 (Y.S.), and T2394484 (Y.S.)), Key Research & Development and Promotion of Special Project (Scientific Problem Tackling) of Henan Province (242102211090) (Y.Z.), the China Postdoctoral Science Foun-dation (2023TQ0300 (Y.Z.), and 2023M743171 (Y.Z.)), and the Post-doctoral Fellowship Program of CPSF (GZB20230666) (Y.Z.). A portion of this work is based on the data obtained at BSRF-1W1A. The authors gratefully acknowledge the cooperation of the beamline scientists at the BSRF-1W1A beamline.

## Author contributions

Y. Z., Y. W., S. Z. and Y. S. initiated the research and designed the experiments; Y. Z., M. Z., Z. W., H. L. and Y. H. fabricated perovskite films and performed material characterizations; M. Z. and Y. C. performed crystallographic characterizations; Y. Z., M. Z. fabricated and measured self-erasure photonic cryptographic chips; Y. Z., Y. W., S. Z. and Y. S. analyzed data; Y. Z. wrote the manuscript. L. J., Y. W., S. Z. and Y. S. guided the work. All authors discussed the results and commented on the manuscript.

## Competing interests

The authors declare no competing interests.
