## [Transparent Peer Review file · Nature Communications]

Direct photo-patterning of halide perovskites toward machine-learning-assisted erasable photonic cryptography

Corresponding Author: Professor Yanlin Song

Version 0:

Reviewer comments:

Reviewer #1

(Remarks to the Author)

This work presents a direct photo-patterning technique for creating erasable patterns in binary mix-halide perovskite films. Leveraging controllable ion migration and photochemical degradation, the method achieves high-fidelity, rapidly self-erasing (within 5 seconds) photoluminescence patterns suitable for photonic cryptography. The speed and erasability are attributed to modulated crystallographic orientation, while neural network-assisted encoding enhances information density and security. This approach offers a significant advancement in the fabrication of erasable photonic devices based on perovskite materials, overcoming limitations of previous complex methods.

1. Segregation in such mixed-halide class of materials for local material modification has been carefully studied previously ["Tracking dynamic phase segregation in mixed-halide perovskite single crystals under two-photon scanning laser illumination" *Small Methods*, 3(11), 1900273 (2019)]. It was shown that under identical illumination, the edges of microcrystals and interior of microplatelets exhibit significantly different phase segregation. How do the authors test such inhomogeneous behavior in their samples?
2. Generally, the segregation is very unpredictable and unstable process, which makes data recording not very reliable. The authors should demonstrate many cycles of writing and erasing. Two-three cycles are not enough for real applications. To the best of my knowledge, self-erased perovskite-based data encryption and reading has been already reported elsewhere ["Photoinduced Transition from Quasi-Two-Dimensional Ruddlesden–Popper to Three-Dimensional Halide Perovskites for the Optical Writing of Multicolor and Light-Erasable Images". *The Journal of Physical Chemistry Letters*, 15(2), 540-548 (2024)]. However, in their work, repeatability has not been proved carefully as well.
3. How does the atmosphere affect the recording and erasing process? I would expect strong influence of humidity on the process.
4. The authors wrote ". Therefore, it is urgent to develop an ideal microscale 63 patterning method for environmentally friendly, high-throughput, high-resolution patterning 64 fabrication of perovskite materials." But is it well-known that Pb-based perovskites are not environmentally friendly platform.
5. There are so many review papers on the topic of laser processing of perovskites for photonic applications with discussions of mechanisms of laser-perovskite interaction. However, the authors do not analyze their results in framework of these works and comparative studies. Why?

Reviewer #2

(Remarks to the Author)

1. As the first diagram in this paper intended to introduce your work, Figure 1a does not clearly show the direct photo patterning process and principle. For example, where is the photo mask used in photolithography? And how does the self-erasure occur from T2 to T3? Regarding Figure 1b, what is the potential meaning of the black substrate and blue pattern?
2. Figures 3a, 3b, and 3d exhibit improper color mapping in the curves and arrows, which may lead to confusion for the reader. The absorption and photoluminescence spectra of the perovskite film undergo a significant redshift with the increased doping concentration of MAI. Taking Figure 3b as an example, the PL peak redshift could be related to the brown curve. What are the differences between the top and bottom pictures in Figure 3i? What reasons could lead to these different

results?

3. Why is the magnification factor in the last photo of Figure 4c different from that in the other photos? Please ensure uniformity in the magnification factor for all PL photographs.

4. What is the highest resolution of this photo-patterning technique? And does this method fit with other micro-nano process technologies, such as photolithography and direct-writing lithography?

5. Please describe the progress of reproducible encryption in detail. Why does the PL photograph appear redder after each cycle? How robustness for this repeatedly encrypt-decrypt system?

6. Figure 5 contains relatively little information but occupies a large space, and there are some repeating elements from Figure 6. Therefore, merging Figures 5 and 6 might be a good idea.

Reviewer #3

(Remarks to the Author)

In this work, the binary mix-halide perovskite films were prepared to realize erasable patterns under UV irradiation. The key materials used for self-erasure photonic pattern encryption were prepared by doping different MAI concentrations to the MAPbBr₃ perovskite films. In order to achieve perovskite thin films with higher ion mobility, the authors introduced BMITFSI additives to obtain 110-oriented MAPbBr₃ perovskite films. Based on these designs, an example of the application of perovskite films in neural network-assisted multi-level photonic pattern encoding and decoding is demonstrated through customized coding rules. In conclusion, the design of this work has certain novelty, especially the preparation of erasable patterns based on perovskite films for information encryption. However, many conclusions are not supported by experiments. Therefore, the following important issues should be well addressed before this manuscript can be recommended for publication.

1. As the authors claimed that, the fabrication of photonic patterns mainly relies on controlled ion migration and degradation of iodine-rich regions within the binary mix-halide perovskite films. It is well known that the main reason for the poor device stability of ionic perovskites is the uncontrollable ion migration. In this work, the authors only obtained perovskite films with high ion migration crystal facet (110) by introducing BMITFSI additives (which has been published), but did not achieve real effective control of ion migration. Therefore, it is recommended that the authors provide effective strategies and methods for the control of ion migration in perovskite films.

2. As shown in Figure 1d, although phase separation or decomposition of mixed halogen perovskite can be induced by UV light irradiation, its controllability and repeatability are worrisome, and the authors are recommended to provide detailed demonstration and discussion.

3. In Figure 2, the authors show that MAPbBr₃ perovskite films with dominant orientation on the 110-crystal plane are obtained by BMITFSI additives. However, the material used for the erasable pattern preparation is the mixed halogen perovskite films rather than MAPbBr₃ perovskite films. It has not been confirmed in this work whether BMITFSI has the same crystallization regulation effect after different concentrations of MAI are doped into MAPbBr₃ perovskite.

4. The lower ion migration activation energy for the (110) crystal facet than the (100) crystal facet in Figure 2n should be further demonstrated by experiments, not only calculation.

5. The degradation mechanism of perovskite MAPbI₃ film under UV irradiation was discussed in Figure 3h. I think this is not support for this work, because the materials ultimately used for photonic pattern coding and decoding applications are mix halide perovskites, whose degradation mechanism is much more complex than that of single halide perovskite MAPbI₃ films. The authors attempt to prove the controllability mechanism of this technique with the perovskite MAPbI₃ is obviously incorrect.

6. As can be seen from Figure 4, the pattern resolution of this photon mode encryption technology is not high, which is difficult to achieve accurate storage and reading of information. It is suggested that the author should accurately determine the resolution of the encryption pattern.

7. The authors claim to have realized neural network assisted multi-level photonic pattern coding and decoding with higher storage data capacity and more security and anti-counterfeiting characteristics by customized coding rules. However, in this manuscript, these are only conceptual designs based on schematic presentations and do not achieve the exaggerated effect of the authors. It is suggested that the author delete some exaggerated concept diagrams that have not been realized yet, so as not to mislead the reader.

8. In the title, "transient" is not recommended, as it is not transient from the experimental evidence presented in the manuscript. It is not correct to exaggerate the results, the important is to show that the work has made new progress realistically.

Version 1:

Reviewer comments:

Reviewer #1

(Remarks to the Author)

The authors have addressed all my comments carefully, and the work can be published in its current form, in my opinion.

Reviewer #2

(Remarks to the Author)

The authors made revision according to reviewers' comments and it could be published in current form.

Reviewer #3

(Remarks to the Author)

The authors have addressed all my concerns and suggestions in this revision, and I now recommend the publication of this revised manuscript. Congratulations!

Response to Reviewer's Comments and Revised Details

Manuscript ID: NCOMMS-24-70063

Response to Reviewer 1:

Comment 1: *This work presents a direct photo-patterning technique for creating erasable patterns in binary mix-halide perovskite films. Leveraging controllable ion migration and photochemical degradation, the method achieves high-fidelity, rapidly self-erasing (within 5 seconds) photoluminescence patterns suitable for photonic cryptography. The speed and erasability are attributed to modulated crystallographic orientation, while neural network-assisted encoding enhances information density and security. This approach offers a significant advancement in the fabrication of erasable photonic devices based on perovskite materials, overcoming limitations of previous complex methods.*

Response to Comment 1:

We sincerely thank Reviewer 1 for his/her insightful comments and constructive suggestions, which are helpful to improve the quality of our manuscript. We have carefully revised our manuscript according to Reviewer 1's comments. We believe that the revised version is suitable for publication in *Nature Communications*.

Comment 2: *Segregation in such mixed-halide class of materials for local material modification has been carefully studied previously [“Tracking dynamic phase segregation in mixed-halide perovskite single crystals under two-photon scanning laser illumination” *Small Methods*, 3(11), 1900273 (2019)]. It was shown that under identical illumination, the edges of microcrystals and interior of microplatelets exhibit significantly different phase segregation. How do the authors test such inhomogeneous behavior in their samples?*

Response to Comment 2:

We are thankful to Reviewer 1 for the comments. The phenomenon of inhomogeneous phase separation between the edges and interior of a single-crystal microplate was also demonstrated in our experiments. For our work, it is based on mix-halide perovskite polycrystalline films. The atomic force microscope (AFM) image shows that the size of the crystal grains within the polycrystalline film is roughly a hundred nanometers, which makes it difficult to observe the phase separation and

degradation of iodine-rich regions using in-situ optical characterization (**Figure R1, R2**). Therefore, we grow the corresponding mix-halide perovskite single crystal with a size of roughly 10 micrometers (**Figure R3**). Under continuous UV irradiation, a clear phase separation process was found, which formed iodine-rich edge regions and bromine-rich interior regions, which is consistent with the reported literature (*Small Methods*. **2019**, *3*, 1900273). Furthermore, we found that increased UV irradiation time leads to the disappearance of the iodine-rich region, which can be attributed to the photodegradation mechanism (**Figure R4**). Under UV illumination, the degradation of the iodine-rich region has also been confirmed by previous literature (*J. Mater. Chem. C*, **2019**, *7*, 9326—9334).

Phase separation and decomposition processes were further demonstrated by the in-situ photoluminescence spectrum (**Figure R5, R6**). In contrast to the photoluminescence spectrum of pure MAPbBr₃ and MAPbI₃ perovskites, the photoluminescence spectrum of the bromide-rich phase and iodine-rich phase within mix-halide perovskite single crystal showing a small PL peak difference, indicating that the composition of the bromide-rich phase and iodine-rich phase is close to the pure MAPbBr₃ and MAPbI₃ perovskites (*Sci. China Chem.* **2017**, *60*, 1367–1376; *ACS Energy Lett.* **2016**, *1*, 290–296). Moreover, with the increased UV illumination time, the photoluminescence peak of the iodine-rich phase within the mix-halide perovskite single crystal shifted from 753 nm to 543 nm, suggesting the decomposition of the iodine-rich phase (**Figure R7**).

Figure R1. Atomic force microscope (AFM) (a) topography image and (b) the height diagrams of spin-coated binary mix-halide perovskite films, revealing the small grains size of roughly hundred nanometers.

Figure R2. Photoluminescence photographs of mix-halide perovskite films with BMITFSI additive under different UV irradiation times. The process of phase separation cannot be captured due to the small size of the crystal grains within the polycrystalline film.

Figure R3. Bright-field photographs of mix-halide perovskite single crystal with a size of roughly 10 micrometers.

Figure R4. Photoluminescence photographs of mix-halide perovskite single crystal under different UV irradiation times, revealing phase separation and decomposition processes.

Figure R5. (a) Photoluminescence spectrum of the bromide-rich phase within mix-halide perovskite single crystal and pure MAPbBr₃ perovskites, showing a slight PL peak difference. (b) Photoluminescence spectrum of the iodine-rich phase within mix-halide perovskite single crystal and pure MAPbI₃ perovskites, showing a small PL peak

difference. The small PL peak difference indicates that the composition of the bromide-rich phase and iodine-rich phase is close to the pure MAPbBr₃ and MAPbI₃ perovskites. Inside images are the corresponding photoluminescence photographs of phase-separated crystals.

Figure R6. Photoluminescence spectrum of the iodine-rich region within mix-halide perovskite single crystal before photodecomposition and after photodecomposition. The shift of the photoluminescence peak from 753 nm to 543 nm suggests the degradation of the iodine-rich phase. Inside images are the corresponding photoluminescence photographs of phase-separated crystals.

To improve our manuscript, we have added the AFM image (**Figure R1**), the bright field photographs of mix-halide perovskite single crystal (**Figure R3**), and the photoluminescence spectrum (**Figures R5, R6**) in the revised Supporting Information as **Fig. S21 (Page 13)**, **Fig. S22 (Page 13)**, **Figs. S23, S24 (Pages 13, 14)**, respectively. We have also replaced **Figure 3g** with the new photoluminescence photographs (**Figure R4**) in the revised manuscript (**Page 10**).

The detailed discussions were added in the main text (please find on **Page 9, Lines 33-34; Page 10, Lines 1-8**) as,

“Because the size of the grains within the polycrystalline film is small, it is difficult to observe the phase separation and decomposition process in the iodine-rich region using in situ optical characterization, so we fabricated mix-halide perovskite single crystals with a size of roughly 10 micrometers (Supplementary Figs. 21, 22). Under continuous UV irradiation, a clear phase separation process was observed with iodine-rich edge regions and bromine-rich interior regions, which is consistent with reported results in the literature⁴⁶. Furthermore, we found the disappearance of the iodine-rich

region with the increased UV irradiation time, which can be attributed to the photodegradation of the iodine-rich phase (Fig. 3g)⁴⁵. Phase separation and decomposition processes were also demonstrated by the photoluminescence spectra (Supplementary Figs. 23, 24).”

The detailed caption changes of Fig. 3 were replaced in the main text (please find on **Page 11, Lines 8-14**) as,

“g. Photoluminescence photographs and schematic diagram of mix-halide perovskite single crystals under different UV irradiation time, revealing phase separation and decomposition of iodine-rich phases. The top picture is the experimental data, and the bottom picture is the corresponding schematic diagram. h, XRD pattern of the mix-halide perovskite film before and after UV irradiation. i, Decomposition mechanism of mix-halide perovskite film under UV irradiation. j, Microscopic ion distribution during phase separation and decomposition of mix-halide perovskite film.”

Furthermore, we have carefully read the literature related to the phase separation and photodegradation of mix-halide perovskites and added related references in the revised manuscript (please find on **Page 20, Lines 9-13**).

“45. Ruan, S. et al. Light-induced degradation in mixed-halide perovskites. *J. Mater. Chem. C* **7**, 9326-9334 (2019).

46. Chen, W., Mao, W., Bach, U., Jia, B. & Wen, X. Tracking Dynamic Phase Segregation in Mixed-Halide Perovskite Single Crystals under Two-Photon Scanning Laser Illumination. *Small Methods* **3**, 1900273 (2019).”

*Comment 3: Generally, the segregation is very unpredictable and unstable process, which makes data recording not very reliable. The authors should demonstrate many cycles of writing and erasing. Two-three cycles are not enough for real applications. To the best of my knowledge, self-erased perovskite-based data encryption and reading has been already reported elsewhere [“Photoinduced Transition from Quasi-Two-Dimensional Ruddlesden - Popper to Three-Dimensional Halide Perovskites for the Optical Writing of Multicolor and Light-Erasable Images”. *The Journal of Physical Chemistry Letters*, 15(2), 540-548 (2024)]. However, in their work, repeatability has not been proved carefully as well.*

Response to Comment 3:

We thank Reviewer 1 for pointing out this issue. In the previous version of our

manuscript, the robustness of this approach is indeed not well demonstrated by the fact that writing and erasing patterns have been only measured three times. To confirm the reliability of data recording, we further performed write and erase cycle experiments for 20 times in the revised manuscript. As shown in **Figure R7**, the cycle stability is reliable and there is no noticeable difference after 20 cycles for the same photonic cryptography chips. The pattern is clearly visible after each cycle and the encryption information will not be damaged, thus further proving the reliability of our method.

On the other hand, the fabrication of perovskite films is a low-cost solution processing (*Nat. Rev. Mater.* **2024**, *9*, 759–761; *Chem. Rev.* **2024**, *124*, 10623–10700; *Nat. Rev. Mater.* **2016**, *1*, 15007). In particular, the expensive electron/hole transport layers and the gold electrodes are not required in this experiment, thus further reducing the fabrication cost of the encryption chip. As a result, encryption chips are generally not used indefinitely. After several tens of information transmissions, new encryption chips are usually fabricated. Overall, the reliable cyclic stability and the low-cost, high-throughput fabrication process provide a guarantee for the practical application of encryption chips.

To improve our manuscript, we have added the encryption stability test (**Figure R7**) in the revised Supporting Information as **Fig. S30 (Page S17)**. The detailed discussions were added in the main text (please find on **Page 13, Lines 17-20**) as,

“To demonstrate the reproducible encryption properties of the photonic cryptographic chip, multiple encryptions, decryptions, and self-erasure are carried out, showing excellent reliability, providing a guarantee for the practical application (Figs. 4e, f, Supplementary Figs. 29, 30, and details shown in Supplementary Note S3).”

Figure R7. Multiple re-encryption of photonic pattern information, confirming excellent reliability. The white numbers represent the cycle times.

Comment 4: *How does the atmosphere affect the recording and erasing process? I would expect strong influence of humidity on the process.*

Response to Comment 4:

We thank Reviewer 1 for the professional comment. As Reviewer 1 suggested, we carried out control experiments under different humidity in the revised manuscript. **Figure R8** shows the photoluminescence spectrum of the mix-halide perovskite films under different humidity for 10h, presenting a slight reduction, indicating good humidity stability of perovskite films. As shown in **Figure R9**, the recording and erasing experiments of encryption patterns under different humidity were also carried out, and all encryption chips exhibited clear patterns without noticeable damage under different humidity for 10h, indicating that humidity has little effect on erasable photonic cryptography chip, which can be attributed to the hydrophobic effect of the BMITFSI additive (*Adv. Funct. Mater.* **2024**, *34*, 2314237). Moreover, the encryption and

decoding of pattern information is a fast process, which significantly reduces the exposure time to moisture in the air and increases the lifetime of the photonic cryptographic chip.

On the other hand, the direct photo-patterning technique is applicable to a series of mixed halide perovskite films, such as 3D perovskites, 2D perovskites, etc (**Figure R10**). Compared to 3D perovskites, 2D perovskites exhibit better environmental stability due to their large organic cations significantly hindering the erosion of humidity (*Nat. Energy* **2024**, *9*, 1540–1550; *Nat. Energy* **2021**, *6*, 63–71; *Nat. Photonics* **2020**, *14*, 154-163). Therefore, some low-dimensional perovskite with stronger humidity stability can also be employed in high-humidity environments. In addition, the chips are low-cost and generally not used indefinitely. After several tens of information transmissions, new encryption chips are usually fabricated.

To improve our manuscript, we have added the humidity stability test (**Figure R8, R9**) in the revised Supporting Information as **Figs. S31, S32 (Page S18)**. The detailed discussions were added in the main text (please find on **Page 13, Lines 20-25**) as,

“It's worth noting that those encryption chips also present remarkable humidity stability, which can be confirmed by the photoluminescence spectrum and the encryption patterns with no obvious difference under different humidity conditions (Supplementary Figs. 31, 32). To further improve the humidity stability of encryption chips, low-dimensional mixed halide perovskite films can be developed.”

Figure R8. (a) The photoluminescence spectrum of the mix-halide perovskite films under different humidity for 10h (temperature: 25–30°C, relative humidity: 20-60%). (b) Molecular structure of BMITFSI additives. The photoluminescence spectrum of the mix-halide perovskite films presents a slight reduction under different humidity for 10h, indicating good humidity stability of perovskite films, which can be attributed to the

hydrophobic effect of the BMITFSI additive.

Figure R9. The recording and erasing process of encryption patterns based on the mix-halide perovskite films under different humidity of (a) 20%, (b) 40%, and (c) 60% (temperature: 25–30 °C, relative humidity: 20-60%). The fast encryption and decoding of pattern information significantly reduces the exposure time to moisture in the air and increases the lifetime of the photonic cryptographic chip.

Figure R10. The pattern encryption based on (a) 2D PEA₂PbBr₄ perovskite films with MAI concentration of 0.1 mg/ml, (b) binary mix-halide 2D PEA₂SnI₂Br₂ perovskite films.

Comment 5: *The authors wrote “Therefore, it is urgent to develop an ideal microscale patterning method for environmentally friendly, high-throughput, high-resolution patterning fabrication of perovskite materials.” But is it well-known that Pb-based perovskites are not environmentally friendly platform.*

Response to Comment 5:

We thank Reviewer 1 for pointing out this issue. We agree with the reviewer that

Pb-based perovskites are not environmentally friendly platforms. We apologize for the misunderstanding. To improve the manuscript, this sentence has been corrected (please find on **Page 3, Lines 27-29**) as,

“Therefore, it is urgent to develop an ideal microscale patterning method for low-cost, high-throughput, high-resolution patterning fabrication of perovskite materials.”

Furthermore, we fabricated lead-free Sn-based perovskites encryption chips. XRD pattern and absorption spectra demonstrate the successful fabrication of binary mix-halide 2D $\text{PEA}_2\text{SnI}_2\text{Br}_2$ perovskite films (**Figure R11**). As shown in **Figure R12**, the encryption chip exhibits good recording and erasing ability, providing an environmentally friendly platform. However, in contrast to Pb-based perovskites, the photoluminescence quantum yields of 3D or 2D lead-free Sn-based perovskites with narrow band emissions are usually low (*Nat. Photon.* **2023**, *17*, 755–760; *J. Am. Chem. Soc.* **2019**, *141*, 10324–10330; *Adv. Mater.* **2018**, *30*, 1706592). To further improve the quality of patterning, binary mix-halide Sn-based perovskite films with higher photoluminescence quantum yields should be developed in the future.

To improve our manuscript, we have added the patterning of binary mix-halide Sn-based $\text{PEA}_2\text{SnI}_2\text{Br}_2$ perovskite films (**Figure R11, R12**) in the revised Supporting Information as **Figs. S33, S34 (Page S19)**. The detailed discussions were added in the main text (please find on **Page 13, Lines 25-27**) as,

“The direct photo-patterning technique has also been used for binary mix-halide Sn-based perovskite films, providing an environmentally friendly platform. (Supplementary Figs. 33, 34).”

Figure R11. (a) The XRD pattern and (b) absorption spectrum of binary mix-halide $\text{PEA}_2\text{SnI}_2\text{Br}_2$ perovskite films.

Figure R12. The recording and erasing process of encryption patterns based on the binary mix-halide $\text{PEA}_2\text{SnI}_2\text{Br}_2$ perovskite films.

Comment 6: *There are so many review papers on the topic of laser processing of perovskites for photonic applications with discussions of mechanisms of laser-perovskite interaction. However, the authors do not analyze their results in framework of these works and comparative studies. Why?*

Response to Comment 6:

We thank Reviewer 1 for the valuable comment. We agree with Reviewer 1 that the discussion of laser processing of perovskites in this paper is indeed scarce. According to Reviewer 1's comment, we have carefully searched for review articles on laser processing of perovskites (*Light Adv. Manuf.* **2024**, 4, 14; *Adv. Opt. Mater.* **2024**, 12, 2302782; *Adv. Mater. Technol.* **2023**, 8, 2200275). Laser processing technology greatly promotes the functionalized application of perovskite optoelectronic devices, but the mechanism of light-matter interaction is complicated. Currently, the main light-matter interaction mechanisms for laser patterning of perovskite can be classified into eight categories: laser ablation, laser-induced crystallization, laser-induced defect-management, laser-induced ion migration, laser-induced phase segregation, laser-induced phase transitions, laser-induced dimensionality change, and laser-induced photoreaction. Based on the above mechanism, the patterning fabrication of perovskites was realized and applied to functional devices, such as photodetectors, lasers, optical information storage, flat lenses, solar cells, and light-emitting diodes (*Adv. Opt. Mater.* **2022**, 10, 2200856; *Nano Lett.* **2021**, 21, 10019-10025; *Science* **2022**, 375, 307-310;

Adv. Mater. **2020**, *32*, 2001388; *Energy Environ. Sci.* **2020**, *13*, 1187-1196; *Sci. Adv.* **2022**, *8*, eabm8433).

In previous manuscripts, less discussion of laser processing was mainly attributed to the following two reasons. First, the innovation of this work is the development of a new direct photo-patterning technique using UV light based on binary mix-halide perovskite films, realizing an erasable photonic cryptography chip with fast self-erasure time within 5 seconds. The entire procedure was realized based on UV light-emitting diodes without the use of lasers, thus ignoring the literature related to laser processing. Second, in this paper, we systematically discuss the regulation of ion migration rate and the mechanism of pattern recording and erasing under UV illumination, proposing a new photochemical reaction mechanism of patterning. Essentially, it is also a discussion of light-matter interaction. In summary, we discuss in detail the mechanism of light-matter interaction during patterning under UV light-emitting diode illumination in this paper, but the discussion of the article on laser processing of perovskite is lacking. Once again, thank you for your valuable comment.

To improve our manuscript, detailed discussions were added in the main text (please find on **Page 3, Lines 15-18**) as,

“Lately, the direct laser writing (DLW) technique was successfully applied to the patterning of perovskite materials, realizing the fabrication of functional optoelectronic devices²⁵⁻²⁹. For example, Gan and his colleagues realized multicolor fluorescence patterns based on the DLW technique, which provides new insights into anticounterfeiting and steganography³⁰.”

We also added related references in the revised manuscript (**Page 18, Lines 31-34; Page 19, Lines 1-9**).

“25. Sheng, Y., Wen, X., Jia, B. & Gan, Z. Direct laser writing on halide perovskites: from mechanisms to applications. *Light Adv. Manuf.* **4**, 14 (2024).

26. Cherepakhin, A. et al. Advanced Laser Nanofabrication Technologies for Perovskite Photonics. *Adv. Opt. Mater.* **12**, 2302782 (2024).

27. Wang, Z. et al. Flat Lenses Based on 2D Perovskite Nanosheets. *Adv. Mater.* **32**, 2001388 (2020).

28. Anoshkin, S.S. et al. Photoinduced Transition from Quasi-Two-Dimensional Ruddlesden–Popper to Three-Dimensional Halide Perovskites for the Optical Writing of Multicolor and Light-Erasable Images. *J. Phys. Chem. Lett.* **15**, 540–548 (2024).

29. Wan, Z., Liu, Z., Zhang, Q., Zhang, Q. & Gu, M. Laser Technology for Perovskite: Fabrication and Applications. *Adv. Mater. Technol.* **9**, 2200275 (2024).
30. Ma, K. et al. Tunable Multicolor Fluorescence of Perovskite-Based Composites for Optical Steganography and Light-Emitting Devices. *Research* **2022**, 9896548 (2022).”

Response to Reviewer 2:

Comment 1: *As the first diagram in this paper intended to introduce your work, Figure 1a does not clearly show the direct photo patterning process and principle. For example, where is the photo mask used in photolithography? And how does the self-erasure occur from T2 to T3? Regarding Figure 1b, what is the potential meaning of the black substrate and blue pattern?*

Response to Comment 1:

We thank Reviewer 2 for the professional comment. We apologize for the misunderstanding regarding the schematic diagram of **Figs. 1a, 1b**. First, we re-modified the schematic to more clearly show the position of the photo mask and the process of self-erasure (**Figure R13a**). Self-erasure of pattern information occurs simultaneously with the reading of the pattern information.

Furthermore, we explain the potential meaning of the black background with a blue pattern. To validate the universal applicability of the direct photo-patterning technique for different types of binary mix-halide perovskite films, the patterning of binary mix-halide 2D perovskite films was also fabricated, presenting a crimson background with a blue pattern. Therefore, the schematic of the crimson background with blue patterns is better aligned with the experimental results, corresponding to the patterning of binary mix-halide 2D perovskite films (**Figure R13b**).

To improve our manuscript, we have replaced **Figs. 1a, 1b** with the new schematic diagram (**Figure R13**) in the revised manuscript (**Page 4**). We also added the discussion in the revised manuscript (**Page 5, Line 18-21**):

“Patterned information is read under UV illumination, while its pattern information is hidden in the bright field image. The patterned information is erased simultaneously with the reading of the information, thus avoiding the secondary leakage of information (Figs. 1a, 1c, and Supplementary Fig. 1).”

The detailed caption of **Figs. 1a, b** were replaced in the main text (please find on **Page 4, Lines 14-17; Page 5, Lines 1-2**) as,

“a, Schematic diagram and photonic pattern based on direct photo-patterning technique. Self-erasure of pattern information simultaneously occurs with the read information. b, The universal pattern fabrication of direct photo-patterning technique for different perovskite compositions, shapes, and sizes. The yellow and crimson

background photoluminescence colors correspond to binary mix-halide 3D and 2D perovskite films, respectively.”

Figure R13. (a) Schematic diagram and photonic pattern based on direct photo-patterning technique. Self-erasure of pattern information simultaneously occurs with the read information. (b) The universal pattern fabrication of direct photo-patterning technique for different perovskite compositions, shapes, and sizes. The yellow and crimson background photoluminescence colors correspond to binary mix-halide 3D and 2D perovskite films, respectively.

Comment 2: *Figures 3a, 3b, and 3d exhibit improper color mapping in the curves and arrows, which may lead to confusion for the reader. The absorption and photoluminescence spectra of the perovskite film undergo a significant redshift with the increased doping concentration of MAI. Taking Figure 3b as an example, the PL peak redshift could be related to the brown curve. What are the differences between the top and bottom pictures in Figure 3i? What reasons could lead to these different results?*

Response to Comment 2:

We thank Reviewer 2 for raising this constructive suggestion. To more clearly show the correlation between the spectral curves and the corresponding arrows, we recalibrated their colors (**Figure R14, R15**).

We apologize for the misunderstanding caused by the unclear caption. The bottom pictures are the schematic diagrams corresponding to the top experimental pictures. To further improve the quality of the manuscript, we have added new captions to describe the **Fig. 3g**.

To improve our manuscript, we have replaced **Figs. 3a, 3b, 3d, 3g** with the new figures (**Figure R14, R15, R16**) in the revised manuscript (**Page 10**).

The detailed caption of **Fig. 3g** was replaced in the main text (please find on **Page 11, Lines 8-11**) as,

“g. Photoluminescence photographs and schematic diagram of mix-halide perovskite single crystals under different UV irradiation time, revealing phase separation and decomposition of iodine-rich phases. The top picture is the experimental data, and the bottom picture is the corresponding schematic diagram.”

Figure R14. (a) The absorption and (b) the photoluminescence spectrum of the MAPbBr₃ perovskite film with the different MAI doping concentrations.

Figure R15. Photoluminescence photographs of mix-halide perovskite films with different MAI doping concentrations under different UV irradiation times.

Figure R16. Photoluminescence photographs of mix-halide perovskite single crystals under different UV irradiation, revealing phase separation and decomposition of iodine-rich phases. The top picture is the experimental data, and the bottom picture is the corresponding schematic diagram.

Comment 3: *Why is the magnification factor in the last photo of Figure 4c different from that in the other photos? Please ensure uniformity in the magnification factor for all PL photographs.*

Response to Comment 3:

We thank Reviewer 2 for the valuable comment. We apologize for our negligence regarding the different magnification factors in **Fig. 4c**. As Reviewer 2 suggested, we have revised the picture (**Figure R17**).

To improve our manuscript, we have replaced **Fig. 4c** with the revised picture (**Figure R17**) in the revised manuscript (**Page 12**).

Figure R17. Schematic and photoluminescence photographs of the decryption and self-erasure process of photonic pattern information.

Comment 4: *What is the highest resolution of this photo-patterning technique? And does this method fit with other micro-nano process technologies, such as photolithography and direct-writing lithography?*

Response to Comment 4:

We thank Reviewer 2 for their valuable comments on the attempt of the highest resolution of the direct photo-patterning technique. To confirm the resolution of the direct photo-patterning technique, we carried out relevant experiments. **Figure R18** shows the patterning array, which exhibits strict alignment, homogeneous size, and precise position. The smallest pattern size is 4 μm , which corresponds to a resolution of roughly 3175 PPI. Furthermore, in contrast to the low-resolution pattern with a yellow background, the background color of the high-resolution pattern presents a greener color, probably originating from two reasons. On the one hand, the spacing of the high-resolution pattern is much smaller, thus a portion of light can still penetrate into the space during UV irradiation, resulting in a small amount of phase separation and degradation phenomena at the space; On the other hand, due to the reduced pattern spacing, the color in the space will be affected by the emitted light from the pattern with an arbitrary emission angle.

For photolithography and direct-writing lithography, which are theoretically the same with our direct photo-patterning technique, which are both induced by UV light to produce patterning, thus our techniques are compatible with these two methods. Furthermore, we also carried out corresponding experiments for verification. As shown in **Figures R19, R20**, clear patterns can also be obtained by direct-writing lithography and photolithography based on binary mix-halide perovskite films.

To improve our manuscript, we have added the highest resolution of patterning (**Figure R18**) in the revised Supporting Information as **Fig. S26 (Page S15)**. We have added the photoluminescence pattern based on direct-writing lithography and photolithography (**Figure R19, R20**) in the revised Supporting Information as **Figs. S38, S39 (Page S21, S22)**.

The detailed discussions were added in the main text (please find on **Page 13, Lines 10-12; Page 14, Lines 15-17**) as,

“The resolution of patterning can reach up to 3175 PPI, which provides the feasibility of high-density data storage (Supplementary Figs. 26).”

“Furthermore, clear encryption patterns can also be obtained by photolithography and direct-writing lithography, providing multiple fabrication routes for pattern information encryption (Supplementary Figs. 38, 39).”

Figure R18. Photoluminescence photographs of photonic pattern information based on direct photo-patterning technique, showing the smallest pattern size of 4 μm , corresponding to a resolution of roughly 3175 PPI.

Figure R19. (a) Schematic of the direct-writing lithography technique. (b) Photoluminescence photographs of photonic pattern information based on direct-writing lithography technique.

Figure R20. (a) Schematic of the photolithography technique. (b) Photoluminescence photographs of photonic pattern information based on the photolithography technique.

Comment 5: *Please describe the progress of reproducible encryption in detail. Why does the PL photograph appear redder after each cycle? How robustness for this repeatedly encrypt-decrypt system?*

Response to Comment 5:

(1) Please describe the progress of reproducible encryption in detail.

We thank Reviewer 2 for pointing out this issue. To increase the readability of the manuscript, we have added a detailed reproducible encryption process in the revised Supporting Information as **Supplementary Note 3 (Page S17)**,

“For the fabrication of binary mix-halide perovskite films, MAI of 0.1mg/ml concentration dissolved in an isopropanol solution was spin-coated onto the surface of the MAPbBr₃ perovskite film for an ion-exchange reaction. After the first encryption and erasure, pure isopropanol solution was again spin-coated onto the surface of the binary mix-halide perovskite films, thus causing a redistribution of halide ions to compensate for the loss of ions in the previously exposed regions, and then the second encryption and erasure was performed. The multiple repetitive encryption process is consistent with the above process for reproducible encryption. In particular, since a small amount of iodide ions is taken away during the spin-coating of pure isopropanol solution, thus we will again spin-coat MAI of 0.1 mg/ml concentration onto the surface of the binary mix-halide perovskite films after the fifth cycle to replenish the loss of ion.”

(2) Why does the PL photograph appear redder after each cycle? How robustness for this repeatedly encrypt-decrypt system?

We thank Reviewer 2 for the professional comment. To verify if the PL photograph color becomes redder after each cycle, we re-perform the cyclic stability experiments for 20 times. As shown in **Figure R21**, after 20 cyclic stability experiments, the background color of the encryption information does not gradually become redder with the increasing number of cycles. The pattern is clearly visible after each cycle, providing a guarantee for the practical application of the encryption chip. In previous manuscripts, color differences of PL photographs may be originated from the following two reasons. First, color differences perhaps come from setting variations of the fluorescent microscope, such as exposure, gain, brightness, white balance, etc. Second, the color difference may stem from the inhomogeneous exchange of ions at different regions within the encryption chip, since the ion exchange process is easily affected by environmental temperatures and other conditions. In summary, although there are slight background and pattern color differences during repetitive encryption, these differences will not affect the information encryption, decryption, and erasable processes, mainly

because the process of encrypting and transmitting information relies on the pattern within the encryption chip.

To improve our manuscript, we have added the cyclic stability experiments (**Figure R21**) in the revised Supporting Information as **Fig. S30 (Page S17)**. We have replaced **Figs. 4e, 4f** with the revised picture (**Figure R22**) in the revised manuscript (**Page 12**). The detailed discussions were added in the main text (please find on **Page 13, Lines 17-20**) as,

“To demonstrate the reproducible encryption properties of the photonic cryptographic chip, multiple encryptions, decryptions, and self-erasure are carried out, showing excellent reliability, providing a guarantee for the practical application (Figs. 4e, f, Supplementary Figs. 29, 30, and details shown in Supplementary Note S3).”

Figure R21. Multiple re-encryption of photonic pattern information, confirming excellent reliability. The white number represents the cycle times.

Figure R22. Multiple re-encryption of photonic pattern information, confirming excellent reliability. The figures are extracted from the Supplementary Fig. 30. The white number represents the cycle times.

Comment 6: *Figure 5 contains relatively little information but occupies a large space, and there are some repeating elements from Figure 6. Therefore, merging Figures 5 and 6 might be a good idea.*

Response to Comment 6:

We thank Reviewer 2 for their valuable comments. According to Reviewer 2's suggestion, we have merged the elements of **Fig. 5** and **Fig.6** in the revised manuscript.

To improve our manuscript, we have replaced **Fig. 5** with the revised picture (**Figure R23**) in the revised manuscript (**Page 15**).

Figure R23. Self-erasure multilevel pattern encoding. (a) The design principle of self-erasure multi-level pattern encryption chip. (b) Photonic pattern encoding based on QR code. (c) Encryption, decryption, and self-erasure process of photonic pattern encoding for "ZZU". (d-f) Neural network-assisted image recognition for photonic pattern decoding.

Response to Reviewer 3:

Comment 1: *In this work, the binary mix-halide perovskite films were prepared to realize erasable patterns under UV irradiation. The key materials used for self-erasure photonic pattern encryption were prepared by doping different MAI concentrations to the MAPbBr₃ perovskite films. In order to achieve perovskite thin films with higher ion mobility, the authors introduced BMITFSI additives to obtain 110-oriented MAPbBr₃ perovskite films. Based on these designs, an example of the application of perovskite films in neural network-assisted multi-level photonic pattern encoding and decoding is demonstrated through customized coding rules. In conclusion, the design of this work has certain novelty, especially the preparation of erasable patterns based on perovskite films for information encryption. However, many conclusions are not supported by experiments. Therefore, the following important issues should be well addressed before this manuscript can be recommended for publication.*

Response to Comment 1:

We thank Reviewer 3 for his/her time and efforts in examining our work. We have addressed all the comments and carefully revised the manuscript according to the insightful suggestions to more clearly present our work. New discussion and experimental results have been added to the revised manuscript and supporting information. We believe that the revised version is suitable for publication in *Nature Communications*.

Comment 2: *As the authors claimed that, the fabrication of photonic patterns mainly relies on controlled ion migration and degradation of iodine-rich regions within the binary mix-halide perovskite films. It is well known that the main reason for the poor device stability of ionic perovskites is the uncontrollable ion migration. In this work, the authors only obtained perovskite films with high ion migration crystal facet (110) by introducing BMITFSI additives (which has been published), but did not achieve real effective control of ion migration. Therefore, it is recommended that the authors provide effective strategies and methods for the control of ion migration in perovskite films.*

Response to Comment 2:

We thank Reviewer 3 for pointing out this issue. We highly agree with the reviewer's comments that poor device stability of ionic perovskites usually originates

from uncontrollable ion migration phenomena, which accelerates the efficiency attenuation of solar cells, increases hysteresis phenomenon of transistors, leads to the unstable electroluminescence and degradation in light-emitting diodes, and so on (*Nat. Mater.* **2022**, *21*, 1396–1402; *Nat. Commun.* **2022**, *13*, 1741; *Adv. Mater.* **2022**, *34*, 2108102). Ion migration phenomenon within perovskite was revealed to be mainly associated with defects of grain boundaries and the crystal faces of perovskites (*Energy Environ. Sci.* **2016**, *9*, 1752-1759; *Angew. Chem. Int. Ed.* **2024**, e202415949). To enhance the performance of optoelectronic devices, ion migration within perovskites needs to be suppressed. Currently, researchers have achieved effective suppression of ion migration in perovskites based on composition engineering (*Angew. Chem. Int. Ed.* **2020**, *59*, 4099 – 4105), dimensional tailoring (*Nat. Commun.* **2021**, *12*, 1246), crystallization modulation (*Nat. Commun.* **2021**, *12*, 361), and surface functionalization (*Nat. Commun.* **2020**, *11*, 170; *Chem. Soc. Rev.* **2023**, *52*, 5516) methods, realizing high-performance optoelectronic devices.

Unlike previously reported optoelectronic devices, our experiments are not aimed to suppress ion migration, but to enhance the rate of ion migration to obtain fast self-erasing photonic encryption devices. In our research, we found that the disappearance of the pattern information requires tens of seconds for binary mix-halide perovskite films without additives, which is not conducive to the fabrication of fast self-erasable photonic encryption devices. Therefore, we further explore methods to improve the erasing speed, which is related to the ion migration rate. Through a series of experiments, we found that the introduction of BMITFSI additives can significantly enhance the self-erasure speed of the pattern. Then, we systematically studied the reason that BMITFSI additives can enhance the erasure speed. First, the introduction of BMITFSI additives can effectively regulate the crystal plane orientation of MAPbBr₃ perovskites and promote the growth of (110) crystal facets. The modulation of crystal plane orientation accompanied by the change of crystal morphology from microplate to polyhedron is an original discovery, which has not been reported in previous literatures (**Figure R24-26**). Furthermore, compared to the (100) crystal facets, the halide ions of the (110) crystal facets possessing lower ion migration activation energies were demonstrated by theoretical simulations and experiments, yielding faster self-erasing photonic encryption devices (**Figure R27**).

Overall, we originally developed an approach to precisely control the crystal facet

orientation of perovskites through the introduction of BMITFSI additives, thus reducing the ion migration activation energy and effectively increasing the ion migration rate (**Figures R27, R28**). By effectively controlling the ion migration rate, an erasable photonic cryptographic chip with a self-erasure time within 5 seconds was obtained.

Figure R24. (a, c) Morphology and (b, d) photoluminescence photographs of MAPbBr₃ perovskite single-crystal fabricated without BMITFSI additives.

Figure R25. (a, c) Morphology and (b, d) photoluminescence photographs of MAPbBr₃ perovskite single-crystal fabricated with BMITFSI additives.

Figure R26. XRD pattern of MAPbBr₃ perovskite single-crystal with/without BMITFSI additives, revealing effective modulation of crystal plane orientation.

Figure R27. (a, b) The calculated ion migration activation energies for the (100) crystal facets and the (110) crystal facets. Temperature-dependent conductivity of the mix-halide perovskite films (c) with BMITFSI additive and (d) without BMITFSI additive.

Figure R28. Photoluminescence photographs of mix-halide perovskite films (a) with BMITFSI additive and (b) without BMITFSI additive under different UV irradiation times.

To improve our manuscript, we have added the morphology, XRD, and temperature-dependent conductivity results (**Figure R24-27**) in the revised Supporting Information as **Figs. S12-15 (Pages S7-S9)**. The detailed discussions were added in the main text (please find on **Page 7, Lines 10-20**) as,

“To explain the differences in ion migration rates for different crystal facets, we systematically measured the ion migration activation energy of perovskite single-crystal with different crystal facets by the temperature-dependent conductivity measurements (details shown in Supplementary Figs. 12-15 and Supplementary Notes S1, S2)^{41,42}. Compared to the E_a value of 0.31 eV of the (100) crystal facet, a lower E_a value of 0.16 eV of the (110) crystal facet was demonstrated, which is close to previously reported literature^{43,44}. Furthermore, the ion migration activation energy on the different crystal facets were calculated based on DFT calculation, which is well in agreement with the experimental value (Figs. 2k-2n, and Supplementary Fig. 16). Overall, in contrast to the (100) crystal facets, the (110) crystal facets have a smaller ion migration activation energy, thus leading to their higher ion migration rate (Figs. 2m, 2n).

We added a detailed fabrication of MAPbBr₃ perovskite single-crystal with/without BMITFSI additives and the calculation of the ion migration activation energy in the revised Supporting Information as **Supplementary Note 1, Note2 (Pages S8-S10)**,

“**Supplementary Note 1.** The fabrication of MAPbBr₃ perovskite single-crystal with/without BMITFSI additives.

The perovskite precursor solution (1.5M) was prepared by dissolving MAPbBr₃

perovskite powder into a solution of DMF. For the perovskite single-crystal without the BMITFSI additive, single crystals were fabricated by dropping a precursor solution onto a glass substrate. Single crystal microplates were formed after evaporation at room temperature. For a solution of perovskite precursor doped with 20% BMITFSI additive (volume ratio relative to perovskite solution), single crystals were fabricated by dropping the precursor solution onto a glass substrate. Polyhedral single crystals were formed after evaporation at room temperature.

Supplementary Note 2. The calculation of the ion migration activation energy of MAPbBr₃ perovskite single-crystal with/without BMITFSI additives.

We then systematically measured the ion migration activation energy of perovskite single-crystal with different crystal facets by temperature-dependent conductivity measurements. The device is based on a lateral photoconductive structure of Au/perovskite/Au. The ion migration activation energy (E_a) can be extracted by fitting the raw data points with the Nernst–Einstein equation,

$$\sigma(T) = \frac{\sigma_0}{T} \exp\left(\frac{-E_a}{k_b T}\right)$$

where k_b is the Boltzmann constant, T is the temperature, and σ_0 is a constant. The ion migration activation energy E_a corresponds to the slope of the $\ln(\sigma T) - 1000/T$ relation. At higher temperature regions, E_a values were extracted from the slope of the fitted line. Supplementary Fig. 15. shows E_a values of the (110) crystal facet and the (100) crystal facet calculated by fitting the corresponding Arrhenius plots. Compared to the E_a value 0.31 eV of the (100) crystal facet, lower E_a value 0.16 eV of the (110) crystal facet was demonstrated.”

We also added related references in the revised manuscript (**Page 19, Lines 33-34; Page 20, Lines 1-8**).

“41. Zhao, Y. et al. Suppressing ion migration in metal halide perovskite via interstitial doping with a trace amount of multivalent cations. *Nat. Mater.* **21**, 1396-1402 (2022).

42. Li, N., Jia, Y., Guo, Y. & Zhao, N. Ion Migration in Perovskite Light-Emitting Diodes: Mechanism, Characterizations, and Material and Device Engineering. *Adv. Mater.* **34**, 2108102 (2022).

43. Meloni, S. et al. Ionic polarization-induced current-voltage hysteresis in CH₃NH₃PbX₃ perovskite solar cells. *Nat. Commun.* **7**, 10334 (2016).

44. McGovern, L., Koschany, I., Grimaldi, G., Muscarella, L.A. & Ehrler, B. Grain Size Influences Activation Energy and Migration Pathways in MAPbBr₃ Perovskite Solar Cells. *J. Phys. Chem. Lett.* **12**, 2423-2428 (2021).”

Comment 3: As shown in Figure 1d, although phase separation or decomposition of mixed halogen perovskite can be induced by UV light irradiation, its controllability and repeatability are worrisome, and the authors are recommended to provide detailed demonstration and discussion.

Response to Comment 3:

We thank Reviewer 3 for pointing out this issue. To confirm the controllability and repeatability of the direct photo-patterning technique, we further performed photonic pattern encryption and erasure cycle experiments in the revised manuscript. As shown in **Figure R29**, the pattern encryption and erasure are reliable and there is no noticeable difference for 10 different photonic cryptography chips, demonstrating excellent controllability and repeatability.

Moreover, cyclic stability experiments were performed for the same photonic cryptography chip for 20 times. There is no noticeable difference after 20 cycles, and the pattern is clearly visible after each cycle and the encryption information will not be damaged (**Figure R30**). Although there are slight background and pattern color differences during repetitive encryption, these differences will not affect the information encryption, decryption, and erasable processes, mainly because the process of encrypting and transmitting information relies on the pattern within the encryption chip.

Overall, the controllability and repeatability of the direct photo-patterning technique is favorable for future practical applications.

To improve our manuscript, we have added the cyclic stability experiments (**Figure R29, 30**) in the revised Supporting Information as **Fig. S29, S30 (Pages S16, S17)**. The detailed discussions were added in the main text (please find on **Page 13, Lines 17-20**) as,

“To demonstrate the reproducible encryption properties of the photonic cryptographic chip, multiple encryptions, decryptions, and self-erasure are carried out, showing excellent reliability, providing a guarantee for the practical application (Figs. 4e, f, Supplementary Figs. 29, 30, and details shown in Supplementary Note S3).”

Figure R29. The pattern encryption and erasure process for 10 different photonic cryptography chips, confirming excellent controllability and repeatability. The white number represents different photonic cryptography chips.

Figure R30. Multiple re-encryption of photonic pattern information, confirming excellent reliability. The white number represents the cycle times.

Comment 4: *In Figure 2, the authors show that MAPbBr₃ perovskite films with dominant orientation on the 110-crystal plane are obtained by BMITFSI additives. However, the material used for the erasable pattern preparation is the mixed halogen perovskite films rather than MAPbBr₃ perovskite films. It has not been confirmed in this work whether BMITFSI has the same crystallization regulation effect after different concentrations of MAI are doped into MAPbBr₃ perovskite.*

Response to Comment 4:

We thank Reviewer 3 for pointing out this issue. As Reviewer 3 suggested, to accurately evaluate the crystallization regulation effect for the perovskite film with different concentrations of MAI, we fabricated binary mix-halide perovskite films and measured the corresponding morphology and XRD. The perovskite films without BMITFSI additive show discrete island-like morphology with large grain sizes, while perovskite films fabricated by the introduction of BMITFSI additives show a denser film morphology with smaller grain sizes, which is consistent with the morphology result of pure MAPbBr₃ perovskite films (**Figure R31, 32**). As shown in **Figure R33, 34**, the MAPbBr₃ perovskite films with different MAI doping concentrations also present stronger (110) diffraction peaks by increasing the proportion of BMITFSI additives, indicating the successful tailoring of the crystal facets. Therefore, the BMITFSI additives have the same crystallization regulation effect for the MAPbBr₃ perovskite films with different concentrations of MAI.

To improve our manuscript, we have added the morphology and XRD results (**Figure R31-34**) in the revised Supporting Information as **Figs. S6-9 (Pages S4-S6)**. The detailed discussions were added in the main text (please find on **Page 7, Lines 1-3**) as,

“Furthermore, the denser film morphology and the same tailoring effect of the crystal facets has also been demonstrated in binary mix-halide perovskite films by the introduction of BMITFSI additives (Supplementary Figs. 6-9).”

Figure R31. Morphology and photoluminescence photographs of mix-halide perovskite films with MAI concentration of 0.1 mg/ml fabricated without (a, b) BMITFSI additive and (c, d) with BMITFSI additive. Perovskite films with additives exhibit smaller crystal domains and a denser structure.

Figure R32. Morphology and photoluminescence photographs of mix-halide perovskite films with MAI concentration of 0.4 mg/ml fabricated without (a, b) BMITFSI additive and (c, d) with BMITFSI additive. Perovskite films with additives exhibit smaller crystal domains and a denser structure.

Figure R33. XRD pattern of MAPbBr₃ perovskite films with the MAI doping concentration of 0.1 mg/ml under different ratios of BMITFSI additives. The perovskite films present the (110)-dominant crystallographic orientation with the increased proportion of BMITFSI additives.

Figure R34. XRD pattern of MAPbBr₃ perovskite films with the MAI doping concentration of 0.4 mg/ml under different ratios of BMITFSI additives. The perovskite films present the (110)-dominant crystallographic orientation with the increased proportion of BMITFSI additives.

Comment 5: *The lower ion migration activation energy for the (110) crystal facet than the (100) crystal facet in Figure 2n should be further demonstrated by experiments, not*

only calculation.

Response to Comment 5:

We agree with Reviewer 3 that lower ion migration activation energy needs to be demonstrated by experiments. In previous manuscripts, we have demonstrated the successful tailoring of the crystal facets by introducing BMITFSI additives, in which the perovskite films present the (110)-dominant crystallographic orientation compared to the (100)-dominant crystallographic orientation using pure solvent. In contrast to perovskite polycrystalline films, perovskite single-crystal without grain boundaries and interfaces enable to reflect more intrinsic ion diffusion and ion migration activation energy of different crystal facets (*Proc. Natl. Acad. Sci. U.S.A.* **2018**, 115 (47), 11929–11934; *ACS Energy Lett.* **2018**, 3, 684–688; *Phys. Chem. Chem. Phys.* **2020**, 22, 11467–11473). Single crystals can also exclude the effect of crystal domain size on the ion migration activation energy. Therefore, perovskite single-crystal with different crystal facet orientations need to be fabricated to more accurately measure the ion migration activation energy.

As shown in **Figure R35, 36**, we first successfully fabricated perovskite single-crystal with different crystal facet orientations by introducing BMITFSI additives. Compared to the microplate morphology of perovskite single-crystals without BMITFSI additives, perovskite single-crystals present a novel polyhedral morphology after the addition of BMITFSI additives. This is a groundbreaking discovery, which has not been reported in the previously published literatures. Furthermore, XRD characterization confirmed the successful modulation of the crystal facet orientations of the single crystals by the introduction of the BMITFSI additive (**Figure R37**). Perovskite single crystals without additives exhibit a (100) crystallographic orientation, while perovskite single crystals with additives exhibit a (110) crystallographic orientation, demonstrating the successful fabrication of single crystals with different crystal facets.

Based on the above results, we then systematically measured the ion migration activation energy of perovskite single-crystal with different crystal facets by the temperature-dependent conductivity measurements (*Nat. Mater.* **2022**, 21, 1396–1402; *Angew. Chem. Int. Ed.* **2023**, 62, e202213932; *Adv. Mater.* **2022**, 34, 2108102.). The device is based on a lateral photoconductive structure of Au/perovskite/Au. The ion

migration activation energy (E_a) can be extracted by fitting the raw data points with the Nernst–Einstein equation,

$$\sigma(T) = \frac{\sigma_0}{T} \exp\left(\frac{-E_a}{k_b T}\right)$$

where k_b is the Boltzmann constant, T is the temperature, and σ_0 is a constant. The ion migration activation energy E_a corresponds to the slope of the $\ln(\sigma T) - 1000/T$ relation. At higher temperature regions, E_a values were extracted from the slope of the fitted line. **Figure R38** shows E_a values of the (110) crystal facet and the (100) crystal facet calculated by fitting the corresponding Arrhenius plots. Compared to the E_a value of 0.31 eV of the (100) crystal facet, a lower E_a value of 0.16 eV of the (110) crystal facet was demonstrated. Smaller ion migration activation energy will lead to a faster ion migration rate. In previous manuscripts, faster color changes for (110)-dominant perovskite film have also been demonstrated under UV irradiation, which can be attributed to smaller ion migration activation energy (**Figure R39**). In conclusion, the (110) crystal facet has lower ion migration activation energy compared to the (100) crystal facet, which has been confirmed by experiments and simulations.

To improve our manuscript, we have added the morphology, XRD, and temperature-dependent conductivity results (**Figures R35-38**) in the revised Supporting Information as **Figs. S12-15 (Page S7-S9)**. The detailed discussions were added in the main text (please find on **Page 7, Lines 10-20; Page 16, Lines 30-33**) as,

“To explain the differences in ion migration rates for different crystal facets, we systematically measured the ion migration activation energy of perovskite single-crystal with different crystal facets by the temperature-dependent conductivity measurements (details shown in Supplementary Figs. 12-15 and Supplementary Notes S1, S2)^{41,42}. Compared to the E_a value of 0.31 eV of the (100) crystal facet, a lower E_a value of 0.16 eV of the (110) crystal facet was demonstrated, which is close to previously reported literature^{43,44}. Furthermore, the ion migration activation energy on the different crystal facets were calculated based on DFT calculation, which is well in agreement with the experimental value (Figs. 2k-2n, and Supplementary Fig. 16). Overall, in contrast to the (100) crystal facets, the (110) crystal facets have a smaller ion migration activation energy, thus leading to their higher ion migration rate (Figs. 2m, 2n).”

“The I - V curves and the temperature-dependent conductivity were measured using

a vacuum manual probe station (Lake Shore) and a 4200 semiconductor characterization system (Keithley, 4200). The device temperature was controlled with a thermoelectric plate and liquid nitrogen.”

We also added a detailed fabrication of MAPbBr₃ perovskite single-crystal with/without BMITFSI additives and the calculation of the ion migration activation energy in the revised Supporting Information as **Supplementary Note 1, Note2 (Pages S8-S10)**,

“Supplementary Note 1. The fabrication of MAPbBr₃ perovskite single-crystal with/without BMITFSI additives.

The perovskite precursor solution (1.5M) was prepared by dissolving MAPbBr₃ perovskite powder into a solution of DMF. For the perovskite single-crystal without the BMITFSI additive, single crystals were fabricated by dropping a precursor solution onto a glass substrate. Single crystal microplates were formed after evaporation at room temperature. For a solution of perovskite precursor doped with 20% BMITFSI additive (volume ratio relative to perovskite solution), single crystals were fabricated by dropping the precursor solution onto a glass substrate. Polyhedral single crystals were formed after evaporation at room temperature.

Supplementary Note 2. The calculation of the ion migration activation energy of MAPbBr₃ perovskite single-crystal with/without BMITFSI additives.

We then systematically measured the ion migration activation energy of perovskite single-crystal with different crystal facets by temperature-dependent conductivity measurements. The device is based on a lateral photoconductive structure of Au/perovskite/Au. The ion migration activation energy (E_a) can be extracted by fitting the raw data points with the Nernst–Einstein equation,

$$\sigma(T) = \frac{\sigma_0}{T} \exp\left(\frac{-E_a}{k_b T}\right)$$

where k_b is the Boltzmann constant, T is the temperature, and σ_0 is a constant. The ion migration activation energy E_a corresponds to the slope of the $\ln(\sigma T) - 1000/T$ relation. At higher temperature regions, E_a values were extracted from the slope of the fitted line. Supplementary Fig. 15. shows E_a values of the (110) crystal facet and the (100) crystal facet calculated by fitting the corresponding Arrhenius plots. Compared to the E_a value

0.31 eV of the (100) crystal facet, lower E_a value 0.16 eV of the (110) crystal facet was demonstrated.”

We also added related references in the revised manuscript (**Page 19, Lines 33-34; Page 20, Lines 1-8**).

“41. Zhao, Y. et al. Suppressing ion migration in metal halide perovskite via interstitial doping with a trace amount of multivalent cations. *Nat. Mater.* **21**, 1396-1402 (2022).

42. Li, N., Jia, Y., Guo, Y. & Zhao, N. Ion Migration in Perovskite Light-Emitting Diodes: Mechanism, Characterizations, and Material and Device Engineering. *Adv. Mater.* **34**, 2108102 (2022).

43. Meloni, S. et al. Ionic polarization-induced current-voltage hysteresis in $\text{CH}_3\text{NH}_3\text{PbX}_3$ perovskite solar cells. *Nat. Commun.* **7**, 10334 (2016).

44. McGovern, L., Koschany, I., Grimaldi, G., Muscarella, L.A. & Ehrler, B. Grain Size Influences Activation Energy and Migration Pathways in MAPbBr_3 Perovskite Solar Cells. *J. Phys. Chem. Lett.* **12**, 2423-2428 (2021).”

Figure R35. (a, c) Morphology and (b, d) photoluminescence photographs of MAPbBr_3 perovskite single-crystal fabricated without BMITFSI additives.

Figure R36. (a, c) Morphology and (b, d) photoluminescence photographs of MAPbBr_3 perovskite single-crystal fabricated with BMITFSI additives.

Figure R37. XRD pattern of MAPbBr_3 perovskite single-crystal with/without BMITFSI additives, revealing effective modulation of crystal plane orientation.

Figure R38. Arrhenius plots of the conductivity for (a) (110) crystal facet with BMITFSI additive, and (b) (100) crystal facet without BMITFSI additive devices under dark conditions, where the E_a is calculated from the slope of the curves.

Figure R39. Photoluminescence photographs of mix-halide perovskite films (a) with BMITFSI additive and (b) without BMITFSI additive under different UV irradiation times.

Comment 6: *The degradation mechanism of perovskite MAPbI_3 film under UV irradiation was discussed in Figure 3h. I think this is not support for this work, because the materials ultimately used for photonic pattern coding and decoding applications are mix halide perovskites, whose degradation mechanism is much more complex than that of single halide perovskite MAPbI_3 films. The authors attempt to prove the controllability mechanism of this technique with the perovskite MAPbI_3 is obviously incorrect.*

Response to Comment 6:

We are thankful to Reviewer 3 for the professional comments. We agree with the reviewer that the degradation mechanism of mixed halide perovskite films is much more complex than that of single halide perovskite MAPbI_3 films. To more clearly

confirm the degradation mechanism in mixed halide perovskite films under UV illumination, we systematically in-situ characterized the degradation process of mixed halide perovskites. In our manuscript, direct photo-patterning technique is based on mix-halide perovskite polycrystalline films with the size of the crystal grains roughly a hundred nanometers, which makes it difficult to observe the phase separation and degradation process of iodine-rich regions using in-situ optical characterization (**Figure R40, R41**). Therefore, we grow the corresponding mix-halide perovskite single crystal with a size of roughly 10 micrometers (**Figure R42**). As shown in **Figure R43**, under continuous UV irradiation, a clear phase separation process was found, which formed iodine-rich edge regions and bromine-rich interior regions, which is consistent with the reported literature (*Small. Methods.* **2019**, *3*, 1900273). Furthermore, we found that increased UV irradiation time leads to the disappearance of the iodine-rich region, which can be attributed to the photodegradation of the iodine-rich region (*J. Mater. Chem. C*, **2019**, *7*, 9326—9334).

Phase separation and decomposition processes were further demonstrated by the in-situ photoluminescence spectrum (**Figure R44, R45**). In contrast to the photoluminescence spectrum of pure MAPbBr₃ and MAPbI₃ perovskites, the photoluminescence spectrum of the bromide-rich phase and iodine-rich phase within mix-halide perovskite single crystal shows a small PL peak difference, indicating that the composition of the bromide-rich phase and iodine-rich phase is close to the pure MAPbBr₃ and MAPbI₃ perovskites (*Sci. China Chem.* **2017**, *60*, 1367–1376; *ACS Energy Lett.* **2016**, *1*, 290–296.). Moreover, with the increased UV illumination time, the photoluminescence peak of the iodine-rich phase within the mix-halide perovskite single crystal shifted from 753 nm to 543 nm, suggesting the photodecomposition of the iodine-rich phase (**Figure R45**).

Furthermore, we systematically investigated the photochemical reaction mechanism of the mixed halide perovskite films under light illumination. First, we systematically reviewed the degradation mechanism of iodine-containing perovskite films under light illumination. Although the degradation mechanism has been studied by a large number of literatures, the specific degradation mechanism is currently controversial (*Nat. Energy* **2016**, *2*, 16195; *Adv. Energy Mater.* **2021**, *11*, 2002326; *Nat. Commun.* **2017**, *8*, 15218; *J. Mater. Chem. C*, **2019**, *7*, 9326—9334.). At present, there are two main degradation mechanisms, iodine-induced photodegradation, and oxygen-

induced photodegradation, detailed below. (i) Iodine-induced photodegradation mechanism: Under UV light exposure, iodide ions will react to form $I_2^{\bullet-}$, which in turn will react with mobile $CH_3NH_3^+$ ions within perovskite films, generating CH_3NH_2 , I_2 , and H_2 . The by-product of I_2 will further accelerate the degradation of perovskite (*Nat. Energy* **2016**, *2*, 16195). (ii) Oxygen-induced photodegradation mechanism: The oxygen diffuses into the iodine vacancies to form superoxide ($O_2^{\bullet-}$) species under UV light exposure, which in turn will react with mobile $CH_3NH_3^+$ ions within perovskite films, generating PbI_2 , H_2O , I_2 , and CH_3NH_2 (*Nat. Commun.* **2017**, *8*, 15218). Overall, the deprotonation reaction of the $CH_3NH_3^+$ ions is the main reason for the photodegradation of iodine-containing perovskite films.

Then, we experimentally studied the degradation process of mix-halide perovskite films. The deprotonation process of the $CH_3NH_3^+$ ions and the decomposition process of mix-halide perovskite films were evidenced by Fourier Transform Infrared spectra (FTIR) and XRD characterization. As shown in **Figure R46**, the N-H stretch vibration and C-H bend vibration intensity within the mix-halide perovskite films are noticeably reduced after UV illumination, revealing a deprotonation reaction of the $CH_3NH_3^+$ ions, which is consistent with previously reported results (*Nat. Commun.* **2020**, *11*, 4172 (2020); *Adv. Energy Mater.* **2021**, *11*, 2002326.). The attenuated diffraction peak intensity of the (110) crystal facet and the new diffraction peak appearance of lead iodide after UV irradiation further indicates the decomposition of mix-halide perovskite films (**Figure R47**). The diffraction peaks shifting to a larger diffraction angle also indicates the decomposition of the iodine-rich phase, which leaves a more stable bromine-rich phase perovskite.

In summary, the photodegradation mechanism of mix-halide perovskite films is complex, and the specific degradation mechanism is still a scientific debate. The deprotonation reaction of the $CH_3NH_3^+$ ions is the main reason for the degradation of mix-halide perovskite films (**Figure R48**).

To improve our manuscript, we have added the AFM image (**Figure R40**), the bright field photographs of mix-halide perovskite single crystal (**Figure R42**), photoluminescence spectrum (**Figures R44, R45**), and FTIR spectra (**Figure R46**) in the revised Supporting Information as **Fig. S21 (Page S13)**, **Fig. S22 (Page S13)**, **Figs. S23, 24 (Page S13, S14)**, and **Fig. S25 (Page 14)**, respectively. We have added the XRD pattern (**Figure R47**) in the revised manuscript as **Fig. 3h (Page 10)**. We also replaced

Figs. 3g, 3i with the new photoluminescence photographs and diagram of the degradation mechanism (**Figure R43. 48**) in the revised manuscript (**Page 10**).

The detailed discussions were added in the main text (please find on **Page 9, Lines 33-34; Page 10, Lines 1-8; Page 11, Lines 15-28**) as,

“Because the size of the grains within the polycrystalline film is small, it is difficult to observe the phase separation and decomposition process in the iodine-rich region using in situ optical characterization, so we fabricated mix-halide perovskite single crystals with a size of roughly 10 micrometers (Supplementary Figs. 21, 22). Under continuous UV irradiation, a clear phase separation process was observed with iodine-rich edge regions and bromine-rich interior regions, which is consistent with reported results in the literature⁴⁶. Furthermore, we found the disappearance of the iodine-rich region with the increased UV irradiation time, which can be attributed to the photodegradation of the iodine-rich phase (Fig. 3g)⁴⁵. Phase separation and decomposition processes were also demonstrated by the photoluminescence spectra (Supplementary Figs. 23, 24).”

“Finally, we elucidate the underlying mechanism for the photodegradation of perovskite films using Fourier Transform Infrared spectra (FTIR) and XRD characterization. As shown in Supplementary Fig. 25, the N-H stretch vibration and C-H bend vibration intensity within the mix-halide perovskite films are noticeably reduced after UV illumination, revealing the deprotonation reaction of the $\text{CH}_3\text{NH}_3^\pm$ ions, which has been demonstrated in previously literatures^{47, 48}. The attenuated diffraction peak intensity of (110) crystal facet and diffraction peak appearance of lead iodide after UV irradiation further indicate the photodegradation of mix-halide perovskite films (Fig. 3h). The diffraction peaks shifting to a larger diffraction angle also indicates the decomposition of the iodine-rich phase, which leaves a more stable bromine-rich phase perovskite. Under UV irradiation, the decomposition mechanism and the corresponding microscopic ion distribution of mix-halide perovskite film are shown in Fig. 3i, j. Overall, the decomposition of the mix-halide perovskite film can be attributed to the deprotonation reaction of the CH_3NH_3^+ ions under UV irradiation, resulting in the complete decomposition of the perovskite structure.”

The detailed caption changes of Fig. 3 were replaced in the main text (please find on **Page 11, Lines 8-14**) as,

“g. Photoluminescence photographs and schematic diagram of mix-halide perovskite single crystals under different UV irradiation time, revealing phase separation and decomposition of iodine-rich phases. The top picture is the experimental data, and the bottom picture is the corresponding schematic diagram. h, XRD pattern of the mix-halide perovskite film before and after UV irradiation. i, Decomposition mechanism of mix-halide perovskite film under UV irradiation. j, Microscopic ion distribution during phase separation and decomposition of mix-halide perovskite film.”

Furthermore, we have carefully read the literatures related to the phase separation and photodegradation of mix-halide perovskites and added related references in the revised manuscript (**Page 20, Lines 9-17**).

“45. Ruan, S. et al. Light-induced degradation in mixed-halide perovskites. *J. Mater. Chem. C* **7**, 9326-9334 (2019).

46. Chen, W., Mao, W., Bach, U., Jia, B. & Wen, X. Tracking Dynamic Phase Segregation in Mixed-Halide Perovskite Single Crystals under Two-Photon Scanning Laser Illumination. *Small Methods* **3**, 1900273 (2019).

47. Hartono, N.T.P. et al. How machine learning can help select capping layers to suppress perovskite degradation. *Nat. Commun.* **11**, 4172 (2020).

48. Wei, J. et al. Mechanisms and Suppression of Photoinduced Degradation in Perovskite Solar Cells. *Adv. Energy Mater.* **11**, 2002326 (2020).”

Figure R40. Atomic force microscope (AFM) (a) topography image and (b) the height diagrams of spin-coated binary mix-halide perovskite films, revealing the small grains size of roughly hundred nanometers.

Figure R41. Photoluminescence photographs of mix-halide perovskite films with BMITFSI additive under different UV irradiation times. The process of phase separation cannot be captured due to the small size of the crystal grains within the polycrystalline film.

Figure R42. Bright-field photographs of mix-halide perovskite single crystal with a size of roughly 10 micrometers.

Figure R43. Photoluminescence photographs of mix-halide perovskite single crystal under different UV irradiation times, revealing phase separation and decomposition processes.

Figure R44. (a) Photoluminescence spectrum of the bromide-rich phase within mix-halide perovskite single crystal and pure MAPbBr_3 perovskites, showing a slight PL peak difference. (b) Photoluminescence spectrum of the iodine-rich phase within mix-halide perovskite single crystal and pure MAPbI_3 perovskites, showing a small PL peak

difference. The small PL peak difference indicates that the composition of the bromide-rich phase and iodine-rich phase is close to the pure MAPbBr_3 and MAPbI_3 perovskites. Inside images are the corresponding photoluminescence photographs of phase-separated crystals.

Figure R45. Photoluminescence spectrum of the iodine-rich region within mix-halide perovskite single crystal before photodecomposition and after photodecomposition. The shift of the photoluminescence peak from 753 nm to 543 nm suggests the degradation of the iodine-rich phase. Inside images are the corresponding photoluminescence photographs of phase-separated crystals.

Figure R46. FTIR spectra of mix-halide perovskite films before and after UV irradiation.

Figure R47. XRD pattern of the mix-halide perovskite film before and after UV irradiation. The attenuated diffraction peak intensity of the (110) crystal facet and the appearance of lead iodide after UV irradiation indicate the degradation of mix-halide perovskite films. The diffraction peaks shifting to a larger diffraction angle further indicates the decomposition of the iodine-rich phase, which leaves a more stable bromine-rich phase.

Figure R48. Degradation mechanism of mix-halide perovskite film under UV irradiation.

Comment 7: *As can be seen from Figure 4, the pattern resolution of this photon mode encryption technology is not high, which is difficult to achieve accurate storage and reading of information. It is suggested that the author should accurately determine the resolution of the encryption pattern.*

Response to Comment 7:

We sincerely appreciate Reviewer 3's valuable suggestions. To enrich the information content and increase the security of the information based on photonic cryptography chips, high-resolution patterning is indeed important. According to

Reviewer 3's suggestion, we have carried out relevant experiments to confirm the pattern resolution of the direct photo-patterning technique. **Figure R49** shows the patterning array based on high-resolution photomasks, which presents a clear pattern with strict alignment, homogeneous size, and precise position. The smallest pattern size is 4 μm , which corresponds to a resolution of roughly 3175 PPI. Furthermore, in contrast to the low-resolution pattern with a yellow background, the background color of the high-resolution pattern presents a greener color, probably originating from two reasons. On the one hand, the spacing of the high-resolution pattern is much smaller, thus a portion of UV light can still penetrate into the space during UV irradiation, resulting in a small amount of phase separation and degradation phenomena at the spacing of the high-resolution pattern; On the other hand, due to the reduced pattern spacing, the whole color is affected by the emitted light from the pattern with an arbitrary emission angle.

To improve our manuscript, we have added the highest resolution of patterning (**Figure R49**) in the revised Supporting Information as **Fig. S26 (Page S15)**.

The detailed discussions were added in the main text (please find on **Page 13, Lines 10-12**) as,

“The resolution of patterning can reach up to 3175 PPI, which provides the feasibility of high-density data storage (Supplementary Fig. 19).”

Figure R49. Photoluminescence photographs of photonic pattern information based on direct photo-patterning technique, showing the smallest pattern size of 4 μm , corresponding to a resolution of roughly 3175 PPI.

Comment 8: *The authors claim to have realized neural network assisted multi-level*

photonic pattern coding and decoding with higher storage data capacity and more security and anti-counterfeiting characteristics by customized coding rules. However, in this manuscript, these are only conceptual designs based on schematic presentations and do not achieve the exaggerated effect of the authors. It is suggested that the author delete some exaggerated concept diagrams that have not been realized yet, so as not to mislead the reader.

Response to Comment 8:

We thank Reviewer 3 for pointing out this issue. As Reviewer 3 suggested, we have deleted some of the concept pictures and merged **Figure 5** and **Figure 6** in our revised manuscript (**Figure R53**). In the previous manuscript, we have already realized simple self-erasing QR code and multi-level pattern encoding with anti-counterfeiting capability (**Figure R53b, c**). **Figure R50** presents the binary encryption rules for pattern information by strictly controlling the length as well as the width of the photonic pattern. The corresponding experimentally prepared photonic encoding patterns are shown in **Figure R51**. To more clearly show the pattern encoding applications, the real information “ZZU” was encoded and decoded based on the direct photo-patterning technique, demonstrating the feasibility of high-level pattern information coding (**Figure R52, R53c**). Neural network-assisted image recognition is further employed for more accurate and efficient decoding of pattern information (**Figure R53d-f**). Once again, thank you for your valuable comment.

To improve our manuscript, we have replaced **Fig. 5** with **Figure R53** in our revised manuscript (**Page 15**). We have added the binary coding rules (**Figure R50**) and the decoding process (**Figure R52**) in the revised Supporting Information as **Fig. S35, S36 (Page S20)**.

Figure R50. (a) Binary coding rules based on photonic patterns with specific widths and lengths. (b) Conversion mechanism of photonic coding to other coding rules.

Figure R51. Encryption, decryption, and self-erasure process of photonic encoding information.

Binary 	01 01 10 10	01 01 10 10	01 01 01 01
Photonic 	2 2 3 3	2 2 3 3	1 1 1 1
Decoding information 	Z	Z	U

Figure R52. The decoding process of encoding information for “ZZU”.

Figure R53. Self-erasure multilevel pattern encoding. (a) The design principle of self-erasure multi-level pattern encryption chip. (b) Photonic pattern encoding based on QR code. (c) Encryption, decryption, and self-erasure process of photonic pattern encoding for “ZZU”. (d-f) Neural network-assisted image recognition for photonic pattern decoding.

Comment 9: In the title, "transient" is not recommended, as it is not transient from the experimental evidence presented in the manuscript. It is not correct to exaggerate the

results, the important is to show that the work has made new progress realistically.

Response to Comment 9:

We would like to sincerely thank Reviewer 3 for this valuable suggestion. We agree with the reviewer that the “transient” used here is not rigorous. As Reviewer 3 suggested, we have replaced “transient” with “erasable” in the revised manuscript.

To strengthen our manuscript, we have replaced the title of the paper in the revised manuscript (please find on **Page 1, Lines 1-2**),

“Direct photo-patterning of halide perovskites toward machine-learning-assisted erasable photonic cryptography”